# A stabilized tandem antigen chimera that elicits potent malaria transmission-reducing activity

Danton Ivanochko[1], Kazutoyo Miura [2,3], Sophia Hailemariam[1,4], Rashmi Ravichandran[5], Yiting Song[6], Wei-Chiao Huang[6], Rianne Stoter[7], Karina Teelen[7], Geert-Jan van Gemert[7], Elizabeth M. Leaf[5], Sidney Chan[5], Christine Men[5], Anthony Semesi[1], Carol Shiu[1], Randall S. MacGill [8], Carole A. Long [2], Matthijs M. Jore [7], Neil P. King [5], Jonathan F. Lovell [6] & Jean-Philippe Julien [1,4,9,10] ✉

Malaria parasite transmission remains a barrier to elimination since asymptomatic individuals sustain the infectious reservoir. Transmission-blocking vaccine (TBV) candidates targeting *Plasmodium falciparum* (Pf) gametocyte surface proteins Pfs230 and Pfs48/45 have shown promise in clinical trials. Several vaccine candidates have been developed for these antigens, yet it is unclear which elicit the most robust and durable transmission-blocking responses. From structure-function relationships of monoclonal antibodies in complex with both antigens, we report the development of a stabilized tandem antigen chimera (STAC), which presents the most potent epitopes from Pfs230 domain 1 (Pfs230-D1) and Pfs48/45 domain 3 (Pfs48/45-D3) in a single construct, while masking non-functional epitopes using an engineered pseudo-native domain disposition. Iterative structure-guided optimization improved antigen yields and stability, while nanoparticle-based multimerization enhanced the functional transmission-reducing activity elicited by the immunogen in female mice. Immunizations with STAC genetically conjugated to self-assembling protein nanoparticles elicited antibodies with potent transmission-reducing activity comparable or superior to the multimerized Pfs230-D1 and Pfs48/45-D3. These findings establish STAC as a promising next-generation TBV candidate to disrupt malaria transmission and accelerate elimination efforts. More broadly, our results support the engineering of highly ordered and stable multi-domain antigens in a single protein as a strategy for the cost-efficient development of multi-component vaccines.

Malaria remains a global health challenge, disproportionately affecting children in sub-Saharan Africa, with the disease causing 597,000 deaths in 2023[1]. The World Health Organization has recommended two malaria vaccines, RTS,S and R21, for pediatric use against *Plasmodium falciparum* (Pf) malaria; together, they have the potential to

significantly reduce morbidity and mortality in this vulnerable population[2,3]. Despite the anticipated success of RTS,S and R21 in reducing pediatric morbidity by inhibiting the infectious pre-erythrocytic lifecycle stage of the Pf parasite, with preliminary data indicating RTS,S is associated with a 22% reduction (95% CI 3%-36%) in

hospital admissions with severe malaria and a 13% reduction (95% CI 3%-23%) in all-cause mortality[2–4], these vaccines do not target the primary human reservoirs of transmission. Epidemiological studies have demonstrated that older age groups can harbor mature gametocytes, the transmissible form of the parasite. Variability in individual transmission potential is a feature of many infectious diseases, and for malaria, the human infectious reservoir is largely sustained by asymptomatic populations, particularly school-aged children and adults, and individuals with submicroscopic infections could transmit the malaria parasite[5–9]. This sustained parasite circulation from human to mosquito represents a critical challenge in malaria control and elimination efforts. Yet, there is no approved vaccine to combat this parasite life-cycle stage, highlighting the need for interventions that target not just symptomatic malaria but also the hidden transmission reservoir. Given the parasite's complex life cycle and the ability of asymptomatic carriers to sustain transmission, expanding vaccine strategies to include adolescents and adults could enhance the effectiveness of RTS,S, R21, and next-generation vaccines, ultimately strengthening malaria control and elimination efforts.

Epidemiological modeling has indicated transmission disruption may be an essential step to achieving malaria elimination[10,11]. Transmission-blocking vaccines (TBVs) are a promising strategy to disrupt the malaria lifecycle through the inhibition of parasite transmission from human to mosquito. By eliciting antibodies that target parasite proteins essential for gamete fertilization and parasite development within the mosquito midgut, TBVs could substantially reduce or eliminate the onward spread of malaria. Pfs230 and Pfs48/45 are sexual-stage surface proteins essential for Pf transmission and represent the two most advanced gametocyte/gamete TBV candidates in clinical development[12,13]. Both proteins belong to the 6-Cysteine (6-Cys) family, categorized by the presence of Immunoglobulin-like folds containing up to three disulfide bonds and are essential for pre-fertilization gametocyte function[14–17]. Pfs230 is comprised of fourteen 6-Cys domains and binds to membrane-bound Pfs48/45, comprising three 6-Cys domains and a GPI anchor, to enable presentation of the heterodimeric complex at the surfaces of gametocytes and gametes[18,19]. While recombinant full-length Pfs230 protein is not an established TBV antigen, the full-length Pfs48/45 protein produced from insect cells is currently being evaluated in a phase I trial with Matrix-M adjuvant (NCT05400746).

Subunit vaccines are capable of focusing immune responses to the most potent regions of a target protein, and TBVs displaying either Pfs230 domain 1 (Pfs230-D1) or Pfs48/45 domain 3 (Pfs48/45-D3) are two such examples currently undergoing clinical evaluation. The clinical Pfs230-D1 candidate contains D1 and a segment of its N-terminal pro-domain (referred to as D1M, amino acids 542–736), which is chemically cross-linked to the carrier protein exoprotein A (EPA)[20,21]. This Pfs230D1M-EPA subunit vaccine is the most advanced TBV candidate with two completed phase I trials assessing adjuvants AS01 and Matrix-M (NCT02942277 and NCT05135273, respectively), and one phase II trial with adjuvant AS01 (NCT03917654). The challenging biochemical tractability of Pfs48/45-D3 (also known as 6 C, amino acids 291–428) once hampered the development of Pfs48/45-D3-based vaccines; however, several protein engineering strategies have enabled recent clinical evaluations. One such example is the phase I trials evaluating a TBV candidate comprising Pfs48/45-D3 fused to a part of blood-stage GLURP antigen (R0.6C) with Alhydrogel or a combination of Alhydrogel and Matrix-M (NCT04862416)[22]. Despite their potential, emerging phase I/II clinical data for the current subunit-based immunogens indicate that these leading TBVs may face significant challenges in generating consistently high levels of functional antibody responses[22–28]. In addition, seroprevalence of antibodies to Pfs230 and Pfs48/45 indicates that some individuals in endemic regions elicit detectable antibody titers exclusively to one individual antigen but not to the other[29,30], so it is conceivable that an immunogen presenting both Pfs230 and Pfs48/45 could have the advantage of boosting the naturally acquired immunity for a broader diversity of individuals. Therefore, the challenge lies in designing TBVs that provide broad and potent protection while also being viable for widespread deployment in endemic regions. Accordingly, the feasibility of manufacturing cost-effective, multi-antigen TBVs at scale remains a key consideration[31].

Strategies that fuse Pfs230 and Pfs48/45 antigenic domains are emerging. Studies have indicated that Pfs230-D1-elicited antibodies typically require complement activation for maximal transmission-reducing activity (TRA), whereas Pfs48/45-D3 can induce potent, complement-independent antibodies[32–38]. These distinct antibody mechanisms of action suggest that a multi-antigen TBV formulation incorporating both targets could provide superior and more consistent efficacy. Such an emerging design is the Pfs48/45-D3 fused to the Pro-domain of Pfs230 connected by a linker composed of PfCSP major and minor repeats[39–41]. This fusion, ProC6C, adjuvanted with Alhydrogel or a combination of Alhydrogel and Matrix-M, has entered clinical evaluation (ISRCTN13649456, PACTR202201848463189)[27,41]. Advances in reverse vaccinology and structure-guided antigen design continue to enable the development of more stable and immunogenic TBV candidates. For Pfs48/45, two recent studies leveraged structural insights from D3 bound to the highly potent mAb TB31F to rationally design stabilizing mutations that conferred enhanced biochemical tractability, leading to increased potency of the antibody response in preclinical studies[42,43]. Recent structure-activity relationship (SAR) studies examining the human monoclonal antibody responses induced by natural exposure to Pf parasites have detailed the molecular blueprints describing functional transmission-reducing activity against Pfs230-D1 and Pfs48/45-D3[34,37]. Antibody responses of Pfs230D1M-EPA clinical vaccinees have uncovered strong antibody responses to non-functional epitopes on Pfs230-D1, and further SAR characterization has identified a molecular determinant for these non-functional antibodies, which are directed to epitopes buried in the full-length native proteins, but exposed in the Pfs230-D1 fragment vaccine candidate[26,33]. Recent advances in protein engineering are also enabling approaches that expand beyond natural proteins[44,45]. As such, a TBV vaccine design approach that is not merely the fusion of distinct Pfs230 and Pfs48/45 protein domains, but rather the efficient display of potent epitopes and the masking of non-functional ones in newly engineered constructs, is conceivable.

In this study, we report the development of a novel transmission-blocking vaccine candidate utilizing a stabilized tandem antigen chimera (STAC) comprising the primary potent epitopes of two leading vaccine antigens into a single polypeptide that masks non-functional epitopes exposed on isolated domains of subunit immunogens. Through a progressive design series, we incorporate interdomain linker optimization and interface stabilization, resulting in improved expression, thermostability, and potent immunogenicity, as determined by immunization using liposomal displayed nanoparticle immunogens. When formulated on protein nanoparticles, STAC elicited robust antibody responses and superior functional activity compared to individual or mixed antigens. Together, these data exemplify a strategy for engineering an optimized multi-antigen presentation in a single protein to enhance antibody responses.

## Results

### Mixing Pfs230-D1 and Pfs48/45-D3 on adjuvanted liposomes elicits strong transmission-reducing activity

To investigate the functional immunogenicity of Pfs230-D1 and Pfs48/45-D3 as individual antigens and assess the potential benefit of their combination, we utilized the cobalt porphyrin-phospholipid (CoPoP) multimerization platform to display His-tagged antigens. Immunogenic CoPoP liposomes, co-incorporating synthetic monophosphoryl lipid A (PHAD) and saponin QS-21 adjuvants (abbreviated as CPQ)[46]

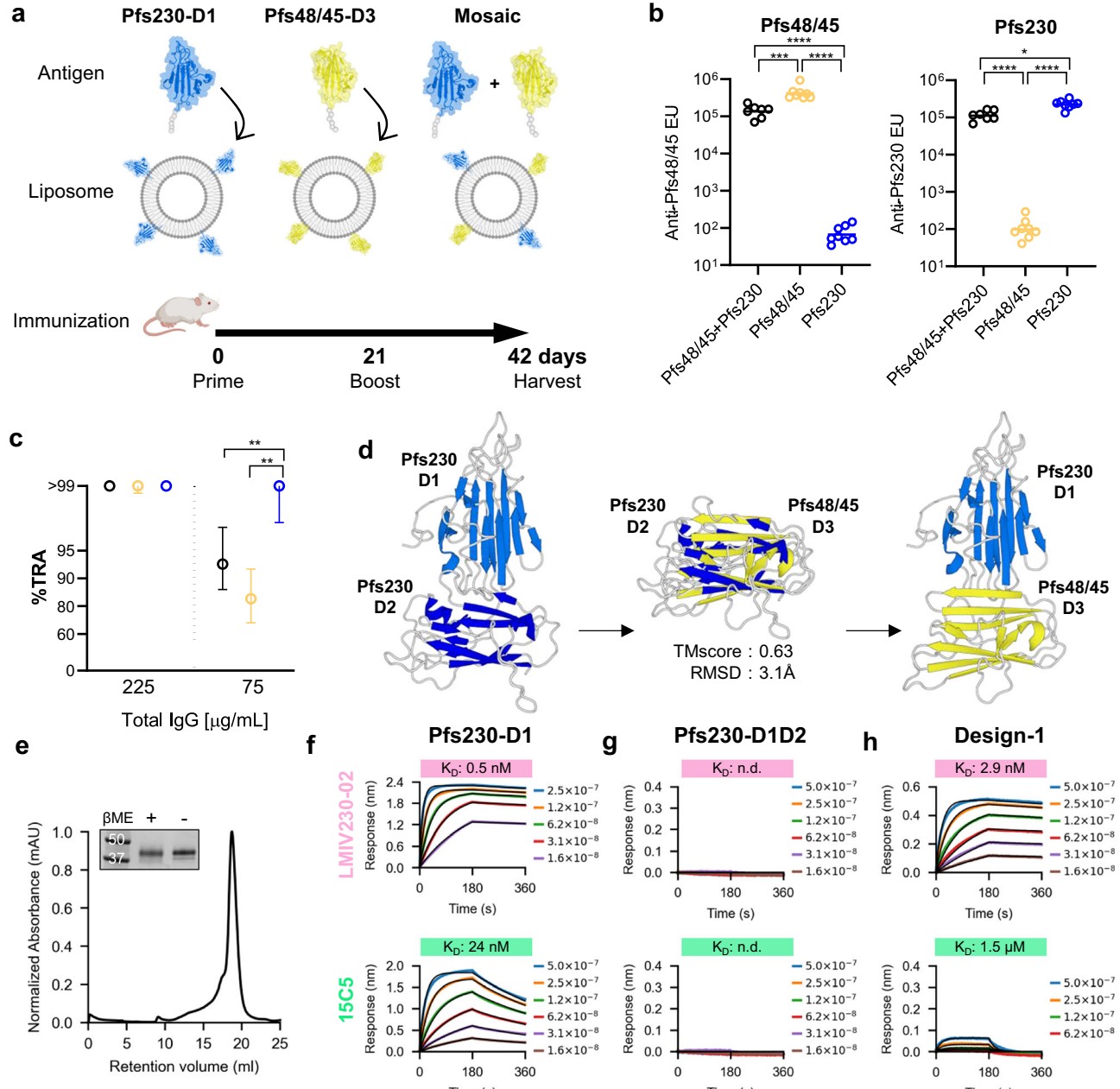

**Fig. 1 | Mixing and coupling potent TBV antigens Pfs230-D1 and Pfs48/45-D3.**
**a** Mouse immunization scheme for antigens on CPQ liposomes (Created with BioRender. Julien, J. (2026) https://BioRender.com/fpyqny8). The individual antigen groups received a 5 μg dose of one of the antigens, while the mosaic group received a 2.5 + 2.5 μg dose. **b** Serum ELISA units (EU) against Pfs48/45 (D3; 6C.mAgE1) and Pfs230 (D1; D1+). Each dot corresponds to individual mice ($n = 8$), and the bar represents the geometric mean. Log-transformed EU values among the three groups were compared by a one-way ANOVA followed by Tukey's multiple comparison tests. *, $p < 0.05$; ***, $p < 0.001$; ****, $p < 0.0001$. **c** SMFA of purified IgG at specified concentrations. The symbols represent the best-estimate percent inhibition in oocyst density (% TRA), with error plotted as a 95% confidence interval. 225 μg/mL data came from a single feed, while 75 μg/mL came from two feeds. The 95% confidence interval and a significant difference among groups were calculated by a zero-inflated negative binomial model, and a Bonferroni correction was made for the p-values. **, $p < 0.01$. Colors correspond to panel (**b**). **d** Design-1 model based off Pfs230-D1D2 (PDBs 7USS and 7ZXF). **e** SEC chromatograph and SDS-PAGE in the presence and absence of β-mercaptoethanol (BME) for Design-1, representative of three independent expressions. Molecular weight markers are indicated in kDa. **f**–**h** BLI binding curves and measured affinities of Pfs230 constructs and Design-1 for non-functional mAbs LMIV230-02 and 15C5. Analyte concentrations (M) are shown. Exact *p*-values available in source data.

were loaded with antigens and used to immunize CD-1 mice in a prime-boost regimen by intramuscular injection. For the Pfs230-D1, we selected the Pfs230D1+ construct (comprising amino acids 552–731), which has been optimized for increased expression and minimal sample heterogeneity by removal of post-translational modification sites, while also presenting less of the N-terminal Pro-domain compared to the clinical construct D1M (amino acids 542–736)[47]. For Pfs48/45-D3, we selected 6C.mAgE1 (amino acids 291–428), which shares the

same domain boundaries as the clinical stage Pfs48/45-D3 antigens and possesses three stabilizing mutations that have been demonstrated to induce superior potency in mouse immunizations compared to the wild-type antigen[42]. Both antigens were expressed in HEK293 cells with C terminal 6x Histidine tags and formulated with CPQ liposomes individually (5.0 μg each) and in mosaic combination (2.5 + 2.5 μg) for immunizations following a 21-day prime-boost regimen (Fig. 1a). By combining Pfs230-D1 and Pfs48/45-D3 on liposomes in this way, we

note that the equal mass amounts used for each antigen resulted in a sub-stoichiometric amount of the larger antigen, Pfs230-D1, to a ratio of 0.74-to-1. At 21-days following the boost immunization, sera were harvested, from which IgGs were purified for evaluation in ELISA and standard membrane feeding assays (SMFA)[48].

As expected, sera from mice immunized with the individual antigens showed strong and specific antibody responses against each antigen (Fig. 1b). Mosaic combination of both antigens showed slightly lower ELISA titers than individual antigens, likely due to the half the dosage of the individual antigens (Fig. 1b). SMFAs using purified IgGs from the mouse immunizations showed that all three antigen groups exhibited nearly complete TRA at 225 μg/ml IgG (Fig. 1c). At a dilution of 75 μg/ml IgG, while all antigen groups exceeded 80% TRA, Pfs230-D1 group showed significantly higher inhibition than the other two groups (Fig. 1c). These data confirmed the compatibility of both Pfs230-D1 and Pfs48/45-D3 antigens to induce biologically active responses when adjuvanted on a multimerizing platform, but also provided a motivation for improved antigen co-display.

### Stacking antigen domains presents epitopes in a pseudo-native arrangement

To reduce complexity, we sought to develop a single antigen capable of presenting potent epitopes from both Pfs230-D1 and Pfs48/45-D3, while simultaneously reducing exposure of non-inhibitory epitopes. Structures of Pfs230-D1D2 and full-length Pfs48/45 have been elucidated, allowing for a structure-based analysis of antigen presentation (Sup Fig. 1a). For both Pfs230 and Pfs48/45, D1 and D2 appear to be rigidly coupled, while flexible linkage of Pfs48/45-D3 appears to allow for a greater degree of conformational freedom[49]. We hypothesized that the presentation of the antigens in a disposition matching the native arrangement of 6-Cys domains may facilitate antigen co-display with structural integrity. Template modeling (TM)-scores[50] were used to cross-compare Pfs230 domains 1 and 2 with Pfs48/45 domains 1, 2, and 3. Domain 1 of each protein gave the highest TM-score (0.71), but intriguingly, Pfs230-D2 displayed the next highest similarity when compared to Pfs48/45-D3 (Sup Fig. 1b). Despite only 18.1% sequence identity between the two domains (Sup Fig. 1c), the TM-score of 0.63 and RMSD of 3.1 Å suggested the possibility that Pfs48/45-D3 may be genetically conjugated to Pfs230-D1 in a manner similar to the native placement of Pfs230-D2 (Fig. 1d). Although this approach would remove Pfs230-D2 from the immunogen, no potent epitopes have been described on this domain.

A theoretical model of Pfs230-D1 tandemly fused to Pfs48/45-D3, herein referred to as Design-1, was constructed by replacing Pfs230-D2 with Pfs48/45-D3 using the domain placement calculated by TMalign on a crystal structure of Pfs230-D1D2 (Fig. 1d). To gauge the theoretical accessibility of potent and non-potent epitopes on Design-1, previously determined crystal structures of antibody-antigen complexes were superimposed on their relevant domains. The epitopes of the potent Pfs230-D1-directed mAbs RUPA-97 and LMIV230-01, as well as the epitopes of the potent Pfs48/45-D3-directed mAbs TB31F and RUPA-44, were found to be fully accessible (Sup Fig. 1d, e). While the epitope of the non-functional mAb RUPA-38 on Pfs230-D1 remained accessible, the epitopes on Pfs230-D1 of the non-functional mAbs LMIV230-02 and 15C5 were sterically occluded by the assumed position of Pfs48/45-D3 (Sup Fig. 1f, g). Unlike RUPA-38, which was identified from an individual naturally exposed to Pf parasites, LMIV230-02 was identified from a study participant immunized with recombinant Pfs230D1M-EPA[26], and 15C5 was generated from a mouse immunized with recombinant Pfs230-D1[51], highlighting how immunization with individually excised domains can elicit non-functional antibodies against epitopes that are not accessible on the gametocyte.

Design-1 was expressed and purified to assess the compatibility of the two tandem domains and to validate the predicted epitope masking. Size-exclusion chromatography indicated that Design-1 predominantly existed as a monomer (Fig. 1e). Binding kinetics experiments confirmed Design-1 maintained epitope fidelity for Pfs230-directed mAbs RUPA-97, LMIV230-01, and RUPA-38, and Pfs48/45-directed TB31F, and RUPA-44 (Sup Fig. 2; Sup Table 1). Binding to LMIV230-02 and 15C5 was assessed for Pfs230-D1, Pfs230-D1D2, and Design-1. As would be expected from their three-dimensional structures, the binding observed for both mAbs to Pfs230-D1 was completely undetected for Pfs230-D1D2 (Fig. 1f, g). For Design-1, binding to LMIV230-02 was partially attenuated by a factor of approximately six-fold, while a greater impact was observed for 15C5 which saw an approximately 60-fold drop in affinity (Fig. 1h). Taken together, these data indicated that while Pfs230-D1:Pfs48/45-D3 fusions are possible to display potent epitopes, an optimal design that masked non-inhibitory epitopes by recapitulating native 6-Cys domain dispositions required further engineering.

### Protein engineering improves the biochemical tractability of stacked antigens

In Design-1, we utilized the inter-domain linker sequence of Pfs230-D1D2 to connect Pfs230-D1 to Pfs48/45-D3; however, partial accessibility of the Pfs230-directed 15C5 and LIMV230-02 epitopes indicated that flexible linkage may have mispositioned the two tandem domains such that the epitopes are not fully occluded. Structural and sequence alignments of Pfs230-D1D2 and Pfs48/45-D2D3 linkers indicated that the backbone positions and amino acid identities are conserved (Fig. 2a, b). To better accommodate the conjugation of Pfs48/45-D3, we hypothesized a shortened and hybrid linker sequence may encourage better stacking of the two domains with the effect of blocking the 15C5 and LMIV230-02 epitopes. We designed six shortened and hybrid linkers, which included an N-to-Q mutation to abrogate a putative N-linked glycan sequon (Fig. 2b). We utilized the potent conformational mAbs RUPA-97 and TB31F that are directed to Pfs230-D1 and Pfs48/45-D3, respectively, to quantify the expression of properly folded, secreted protein designs using BLI. Three of the six designs (Designs −2, −3, and −4) exhibited superior expression relative to Design-1 by BLI quantitation assay (Fig. 2c; Sup Fig. 3a). All three new designs were assessed by SDS-PAGE (Sup Fig. 3b), and Design-4 showed the highest amount of monodisperse species by SEC (Fig. 2d). For Design-4, binding to LMIV230-02 was reduced by a factor of approximately 26-fold compared to Pfs230-D1, while no binding was observed for 15C5 up to 500 nM (Fig. 2e, f), suggesting that Design-4 recapitulated a more desired inter-domain orientation.

To evaluate the structure of Design-4, we determined its co-crystal structure in complex with Fabs RUPA-39 (genetically related and structurally similar to RUPA-97) and RUPA-44 to a resolution of 2.2 Å (Sup Table 2; Sup Fig. 4a–d). Structural alignment with high-resolution crystal structures of Pfs230-D1 and Pfs48/45-D3 demonstrated accurate structural fidelity of each domain with Cα RMSD values of 0.25 Å and 0.30 Å, respectively (Sup Fig. 4e). Similarly, comparison with crystal structures of RUPA-39 bound to Pfs230-D1 (Sup Table 2, Sup Fig. 4f) and RUPA-44 bound to Pfs48/45-D3 confirmed identical binding modes in Design-4, with Cα RMSD values for all heavy chain and light chain CDR residues of 0.52 Å and 0.26 Å, respectively (Sup Fig. 4g). Interestingly, when compared to a crystal structure of Pfs230-D1D2, a dissimilar interdomain orientation was observed for Design-4 (Sup Fig. 4h). When Design-4 was superimposed with the structures of Pfs230-D1 bound by either 15C5 or LMIV230-02, antibody clashing with Pfs48/45-D3 was observed (Sup Fig. 4i, j) even though binding to LMIV230-02 was partially retained. Therefore, while Design-4 demonstrated an improved expression yield, the sub-optimal occlusion of some non-functional epitopes suggested the inter-domain interface may benefit from further stabilization.

The structure of Design-4 provided an opportunity to apply the fixed-backbone sequence design tool ProteinMPNN[44] to generate a new interdomain interface with improved stability and absent binding

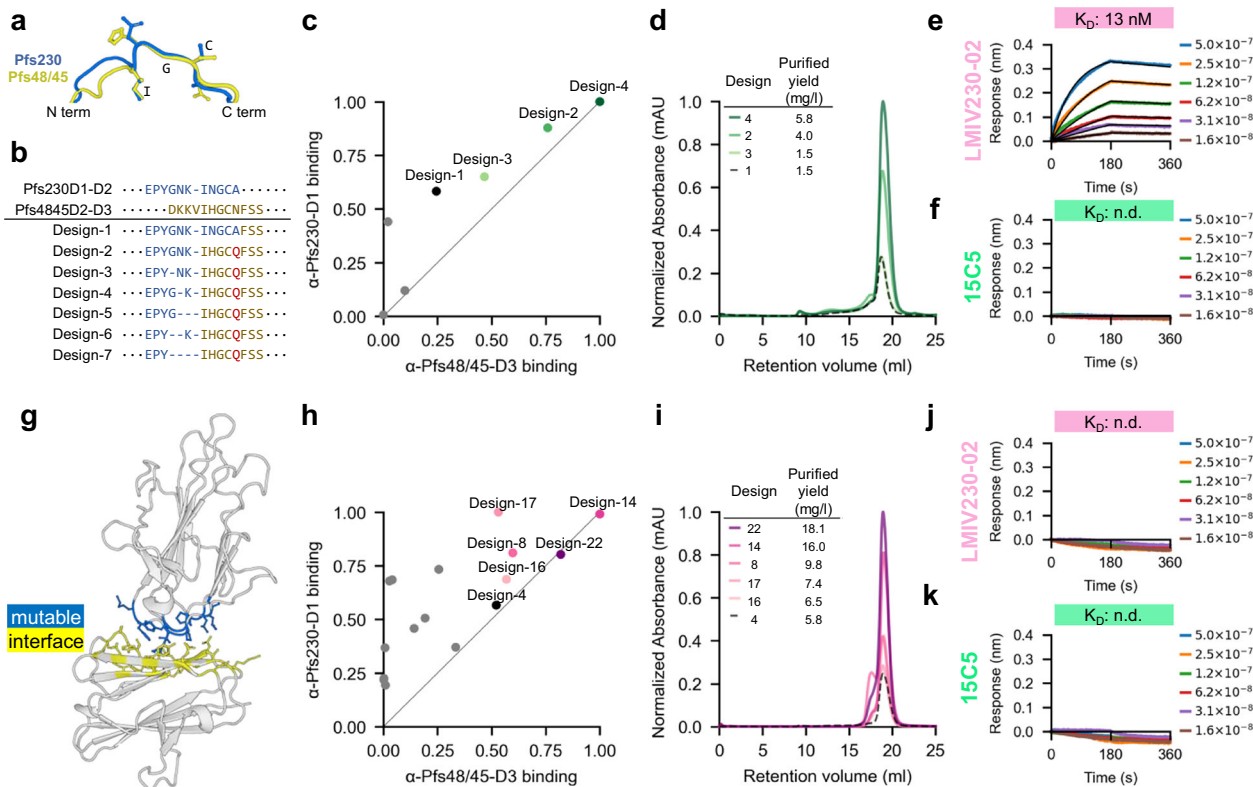

**Fig. 2 | A blended sequence and structural design strategy for improved biochemical tractability. a, b** Overlay of linker structures and sequences for Pfs230-D1D2, Pfs48/45-D2D3, and Design-1 to Design-7. **c** Expression quantitation of antigen designs normalized to the maximum BLI assay response for each antibody. Data points with expression greater than or equal to Design-1 are labeled. **d** Overlayed SEC chromatographs normalized relative to the maximum 280 nm peak with final purified yield data indicated as mg per litre of cell culture. **e, f** BLI curves and binding affinities for Design-4 interaction with nonfunctional mAbs. **g** Crystal structure of Design-4, indicating the mutable interface residues for

ProteinMPNN sequence redesign. **h** Expression quantitation of antigen designs normalized to the maximum BLI assay response for each antibody. Data points with expression greater than or equal to Design-4 are labeled. **i** Overlayed SEC chromatographs normalized relative to the maximum 280 nm peak with final purified yield data indicated as mg per litre of cell culture. **j, k** BLI curves and binding affinities for Design-22 interaction with nonfunctional mAbs. Values for all calculated BLI kinetic parameters corresponding to these plots can be found in Supplementary table 1.

to unwanted mAbs (Fig. 2g). One hundred sequences were generated and filtered using sequence likelihood and recovery metrics, from which 15 diverse sequences were selected for expression testing. Once again, we utilized mAbs RUPA-97 and TB31F, directed to Pfs230-D1 and Pfs48/45-D3, respectively, to quantify the expression of properly folded, secreted protein designs using BLI. Five designs (Designs −8, −14, −16, −17, and −22) exhibited superior expression relative to Design-4 by BLI quantitation assay (Fig. 2h; Sup Fig. 3c). All five new designs were assessed by SDS-PAGE (Sup Fig. 3d), and Design-22 showed the highest expression levels of monodispersed protein by SEC with a yield 12-fold higher than Design-1 (Fig. 2i). Binding kinetics analysis was performed using mAbs RUPA-97, LMIV230-01, RUPA-38, TB31F, and RUPA-44 for Design-4 and Design-22. Binding affinities for all mAbs tested for Design-22 were consistent with those for Pfs230-D1 and Pfs48/45-D3, and within 3-fold of the dissociation constant across all epitopes (Sup Fig. 5; Sup Table 1). Importantly, binding was not detected for either 15C5 or LMIV230-02 (Fig. 2j, k). Here, we identified Design-22 as a stabilized tandem antigen chimera (abbreviated as STAC) that recapitulated the presentation of native potent epitopes, while fully masking known non-potent epitopes on Pfs230-D1.

## Potent epitopes are accurately presented on STAC
To understand the pseudo-native arrangement of Pfs230-D1 and Pfs48/45-D3 in STAC, we performed small-angle X-ray scattering (SAXS) analyses of Designs-1, −4, and STAC, which revealed a reduced radius of gyration (Rg) for both Design-4 and STAC compared to

Design-1 (Sup fig. 6a–c), supporting that the ProteinMPNN-generated sequence recapitulated the solvated structural profile of the template fold. To further elucidate the inter-domain disposition and antigen presentation of the lead design, we determined the structure of STAC bound by two potent Pfs230-D1-directed mAbs, RUPA-97 and LMIV230-01, and by two potent Pfs48/45-D3-directed mAbs, TB31F and RUPA-44, to a resolution of 3.2 Å by cryogenic electron microscopy (cryo-EM) (Fig. 3a; Sup Table 3; Sup Figs. 7, 8). Alignment of STAC with structures of Pfs230-D1 and Pfs48/45-D3 demonstrated accurate structural fidelity for each domain with Cα RMSD values of 0.26 Å and 0.38 Å, respectively (Fig. 3b), despite the incorporation of 16 interface mutations in this new design (Fig. 3c; Sup Fig. 6d).

Similarly, when compared with crystal structures of Pfs230-D1 bound by LMIV230-01 or RUPA-97 and Pfs48/45-D3 with TB31F or RUPA-44, STAC retained identical binding modes in complex with LMIV230-01, RUPA-97, TB31F, and RUPA-44, with combined epitope and paratope Cα RMSD values of 0.35 Å, 0.44 Å, 0.31 Å, and 0.35 Å, respectively (Fig. 3d–k). Notably, STAC exhibited distinct interdomain orientations when compared to crystal structures of Pfs230-D1D2 or Pfs4845-D1D2D3, although in both cases, similar interdomain regions were occluded between either Pfs230-D1 or Pfs48/45-D3 and the adjacent domain (Sup Fig. 6e-f). Interestingly, STAC did not present an identical interdomain orientation when compared to the structure of Design-4, which was used as the input template for the ProteinMPNN interface re-design (Sup Fig. 6g). At a resolution of 3.2 Å, we were able to observe sidechain conformers and recognized a general trend in

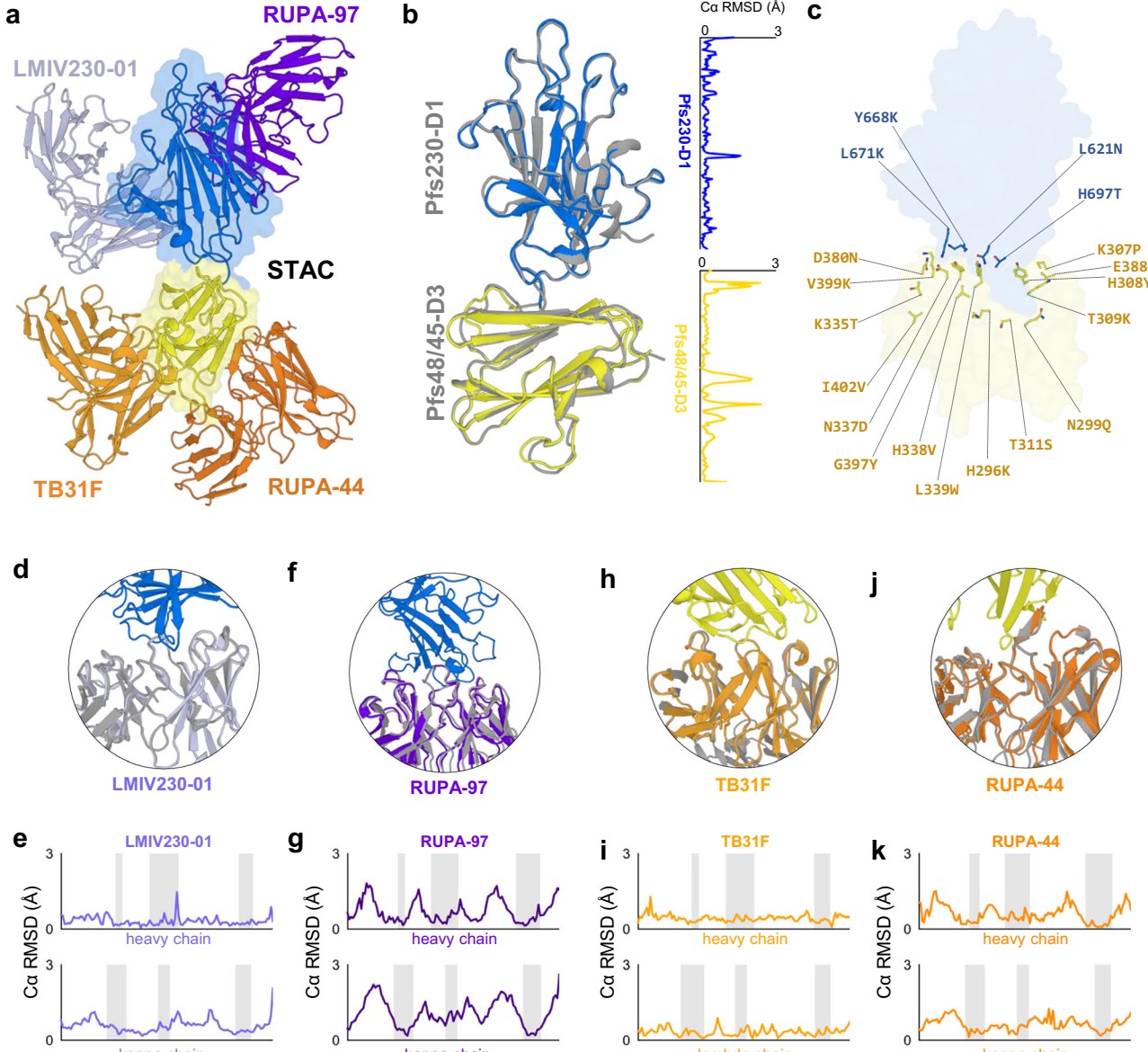

**Fig. 3 | Structural basis for improved stability with intact functional antigenicity. a** STAC bound by Fabs LMIV230-01, RUPA-97, TB31F, and RUPA-44 determined by cryo-EM. **b** STAC structure overlayed with individual Pfs230-D1 and Pfs48/45-D3 domains with Cα RMSD values plotted for each alignment (PDB 9N8N and 7UXL). **c** All mutations involved in stabilization of STAC. The Pfs230 N585Q mutation to ablate a potential *N*-glycosylation site is omitted for clarity. **d–k** Cryo-EM structure of STAC overlayed with previous antigen-bound antibody structures and antibody Cα RMSD values plotted for each alignment (PDBs 7UVQ, 7JUM, 6E63, and 7UXL). Gray coloring indicates previously reported structures when overlayed for comparison. CDR residues (Kabat boundaries) are indicated with shading.

which larger hydrophobic residues packed within the buried interdomain interface, while residues capable of polar interactions were positioned in more solvent accessible regions (Fig. 3c). Collectively, these mutations contributed to the observed increase in monomeric protein expression yield while also concealing the Pfs230-D1 epitopes of 15C5 and LMIV230-02 in a pseudo-native manner due to steric occlusion by the Pfs48/45-D3 domain (Sup Fig. 6h, i).

**Improved stacked antigen stability does not compromise functional immunogenicity**
To assess the immunogenicity of STAC relative to Pfs230-D1, Pfs48/45-D3, and their combination at low dose, we immunized mice with each antigen group loaded onto CPQ nanoparticles (1 µg each) or in combination (0.5 µg + 0.5 µg). While anti-Pfs48/45 ELISA titers were slightly lower in the STAC group compared to Pfs48/45-D3 or the combination groups, anti-Pfs230 ELISA titers were similar among STAC, Pfs230-D1

and the combination groups (Fig. 4a). When the IgGs were tested at 75 µg/mL, the combination, Pfs230-D1, and STAC groups showed almost complete inhibition (>98% inhibition), while Pfs48/45-D3 alone IgG showed 73% inhibition (Fig. 4b). To directly compare the Pfs230-D1 and STAC groups of high functionalities at this low antigen dose, the two IgGs were further tested at four IgG dilutions. Both antigens induced similarly high levels of functional activity, and there was no significant difference between the two IgGs ($p = 0.569$ by a linear regression model) (Fig. 4c). However, the STAC group showed significantly lower anti-Pfs48/45-D3 ELISA titers compared to the Pfs48/45-D3 alone or the combination of Pfs230-D1 and Pfs48/45-D3 groups (Fig. 4a). On CPQ liposomes, STAC is linked to CoPoP by a C terminal 6xHis tag which may bias the response to the apical Pfs230-D1 domain over the basal Pfs48/45-D3 domain. To determine whether antigen orientation on CoPoP influenced immunodominance, we immunized mice with N-tagged STAC (N-STAC) (Sup Fig. 9a). Antibody titers and

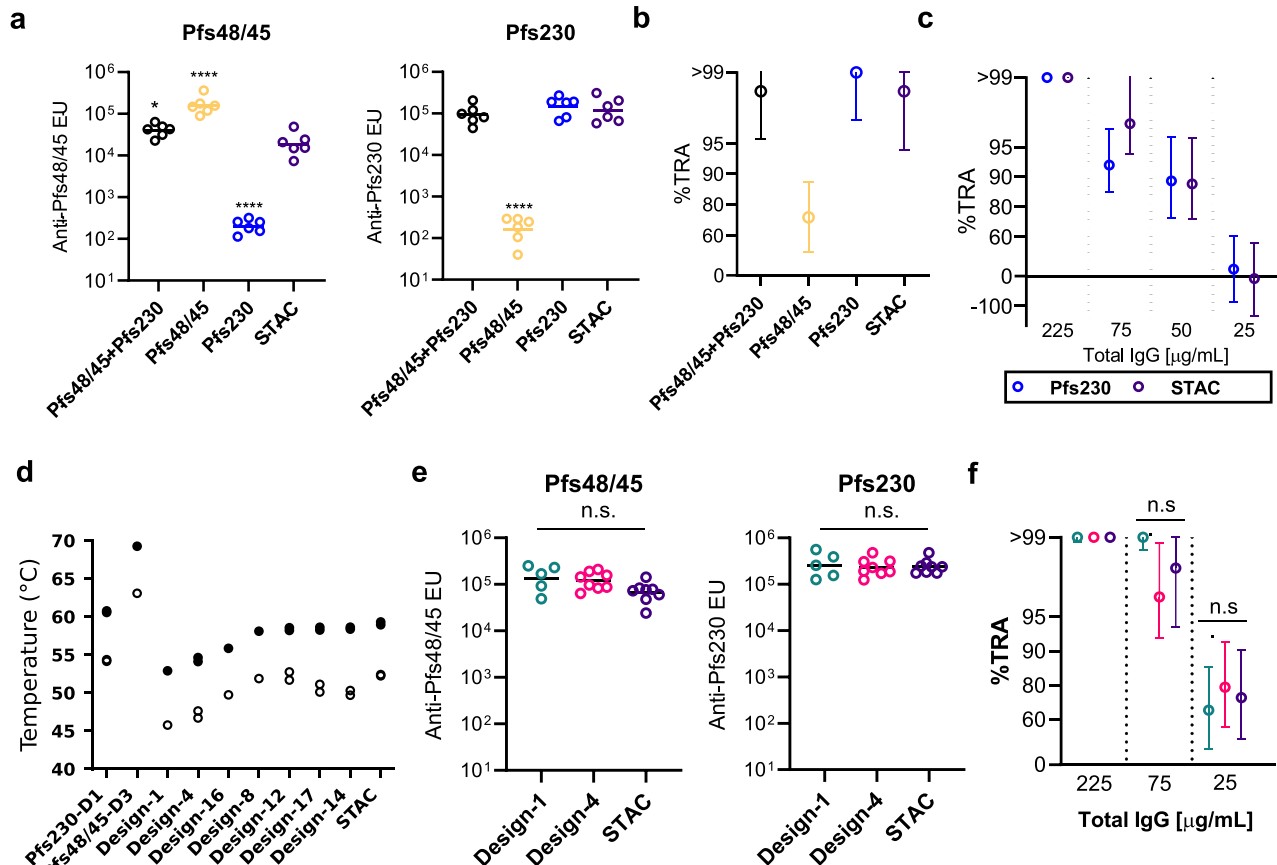

**Fig. 4 | Potent TRA is elicited by vaccination with STAC on CPQ liposomes. a, b** Serum ELISA units (EU) against Pfs48/45 (D3; 6C.mAgE1) and Pfs230 (D1; D1+) and SMFA at 75 μg/ml for all combined (0.5 + 0.5 μg dose) and individual antigens (1 μg dose) compared to STAC (1 μg dose). For the ELISA results, each dot corresponds to individual mice ($n = 6$) and the bar represents the geometric mean. The statistical analyses were performed against the STAC group. *, $p < 0.05$; ****, $p < 0.0001$. **c** SMFA titration for head-to-head comparison of Pfs230-D1 and STAC. One feed was used to generate SMFA data at each IgG concentration, except for 75 μg/mL, which came from two feeds. The symbols represent the best-estimate percent inhibition in oocyst density (% TRA), with error plotted as a 95% confidence interval. **d** Melting temperature ($T_m$; dark circles) and onset values ($T_m^{onset}$; open circles) for

indicated antigen proteins. Two replicates are performed except for Pfs48/45-D3 and Design-8. **e** Serum EU against Pfs48/45 and Pfs230 elicited by Designs −1, −4, and STAC (5 μg dose) on CPQ liposomes. Each dot corresponds to individual mice ($n = 5$, 8, and 8, for Designs −1, −4, and STAC, respectively). n.s.; not significant. **f** SMFA of purified IgG at specified concentrations. The symbols represent the best-estimate percent inhibition in oocyst density (% TRA), with error plotted as a 95% confidence interval. Colors correspond to panel (**e**). For the ELISA data, log-transformed values were compared by one-way ANOVA followed by Tukey's multiple comparison tests. For SMFA data, values were compared by a zero-inflated negative binomial model. Exact *p*-values available in the source data.

functional assays confirmed that N-STAC elicited responses comparable to the C-tagged STAC (C-STAC) (Sup Fig. 9b, c), suggesting a minimal effect of domain orientation on CPQ liposomes for immunodominance in mice.

We previously identified stabilizing mutations in Pfs48/45-D3, incorporated into 6C.mAgE1, enhanced recombinant protein yield and thermostability, both of which were associated with increased potency[42]. In contrast, the development of Pfs230-D1-based immunogens, such as D1M, has not been as adversely affected by poor thermostability[20]. We utilized differential scanning calorimetry (DSC) to assess the thermostability of several tandem antigen chimeras comprising Pfs230-D1 and Pfs48/45-D3, representing the design series starting from Design-1 and leading to STAC. Melting temperatures ($T_m$) and their onset values $T_m^{onset}$ were also measured for the individual domains of Pfs230-D1 (D1+) and Pfs48/45-D3 (6C.mAgE1). We observed $T_m$ and $T_m^{onset}$ values of 60.6 °C and 54.2 °C for Pfs230-D1 and values of 69.2 °C and 63.1 °C for Pfs48/45-D3 (Fig. 4d), which closely matched the previously reported thermostability data for these antigens[20,42]. We found Design-1 possessed the lowest measured $T_m$ and $T_m^{onset}$ values of 52.9 °C and 45.8 °C within the design series. The linker optimization in Design-4 resulted in a minor increase in $T_m$ and $T_m^{onset}$ values to 54.3 °C

and 46.2 °C, while interface re-design further enhanced thermostability with STAC showing the greatest improvement with $T_m$ and $T_m^{onset}$ values of 59.1 °C and 52.3 °C, which most closely approached the parameters observed for Pfs230-D1 (Fig. 4d).

While STAC possessed multiple stabilizing mutations that improved the expression yield and biochemical tractability over both Design-1 and Design-4, whether these alterations would adversely impact functional immunogenicity was undetermined. To address this question, we immunized mice using the same prime-boost and 5 μg dosing regimens as described in Fig. 1 with CPQ liposomes loaded with either Design-1, −4, or STAC antigens. Mouse immunizations with each of these three antigens elicited equivalent high antibody titers against both domains (Fig. 4e) and all resulted in similarly potent antibody responses as determined by SMFA from three antibody dilutions (Fig. 4f), thereby confirming the fidelity of the potent epitopes in the structurally engineered STAC antigen.

## STAC on protein nanoparticles elicits robust transmission-reducing activity

Protein nanoparticles are an established strategy for enhancing the potency of subunit-based vaccines by promoting B cell receptor

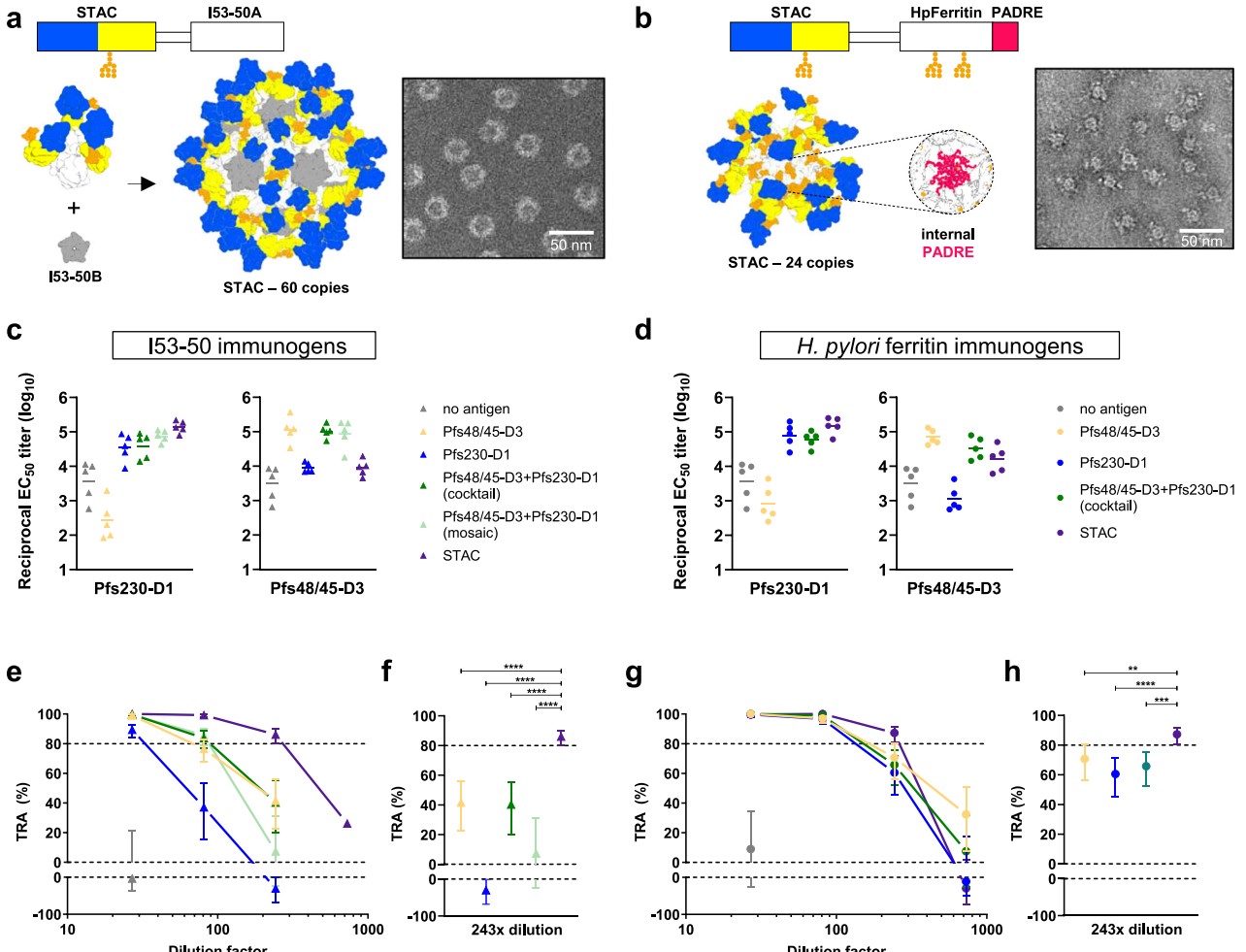

**Fig. 5 | Stabilized tandem antigen chimera on protein nanoparticles elicit robust TRA.** Models of STAC protein nanoparticle immunogens utilizing **a** the two-component I53-50 platform and **b** *H. pylori* ferritin with two engineered N-linked glycans (orange) and the internal PADRE pan-DR T-cell epitope (pink) are shown with nsEM images of each purified protein, representative of two independent experiments. ELISAs from sera of terminal bleeds from **c** I53-50 and **d** *H. pylori* ferritin immunogens. Each dot corresponds to individual mice ($n = 5$) and the bar represents geometric mean titer. SMFA titrations of pooled sera raised against protein nanoparticle immunogens on **e**, **f** I53-50 and **g**, **h** *H. pylori* ferritin. TRA values are estimates from two independent SMFA experiments with oocyst counts for 20 mosquitoes per condition, except for STAC I53-50 at 1:729 dilution which was tested in only one SMFA experiment. Error bars indicate 95% confidence intervals of the TRA estimates. TRA estimates for STAC sera at 1:243 dilution were compared to other sera using a mixed-effects negative binomial regression model, *p*-values: ** <0.01, *** <0.001, **** <0.0001 (**f**, **h**). Colors for SMFA plots correspond to those in the ELISA plots. Immunization regimen and dosages are described in Supplementary fig. 10. Exact *p*-values available in source data.

clustering and processing by antigen-presenting dendritic cells[52,53]. Two clinically validated platforms, *H. pylori* ferritin and I53-50, display 24 and 60 antigen copies, respectively[54,55]. To assess the functional activity elicited by STAC relative to Pfs230-D1 and Pfs48/45-D3 using a multivalent display platform, we conducted an immunization study similar to the previous CPQ liposome experiments but using protein nanoparticles with the adjuvant AddaVax delivered via subcutaneous injections in a prime-boost regimen (Sup Fig. 10a). To ensure comparability across antigen groups and nanoparticle platforms of different valencies, we maintained a consistent molar stoichiometry for each antigen across all immunization groups (Sup Fig. 10b).

For I53-50-based immunogens, STAC, Pfs48/45-D3 (6C.mAgE2[42]), and Pfs230-D1 (D1+[51]) were genetically conjugated to the I53-50A component homotrimer which assembles into nanoparticles in the presence of I53-50B component (Fig. 5a). Self-assembling *H. pylori* ferritin-based immunogens were similarly constructed for each antigen but with two additional nanocage enhancements: engineered N-linked high mannose glycans and the PADRE pan-DR T cell epitope, which have been previously employed for a next-generation PfCSP *H.*

*pylori* ferritin immunogen[56]. Cocktails were generated by mixing independently assembled immunogens of Pfs230 and Pfs48/45 on either I53-50 or *H. pylori* ferritin, while mosaics were formed by co-assembling mixtures of Pfs230 and Pfs48/45 on I53-50A with I53-50B. Importantly, the molar stoichiometries of all cocktail and mosaic groups were twice that of the individual antigen groups to account for the two antigenic domains present in the STAC design (Sup Fig. 10b).

Immunizations elicited robust and specific antibody responses, as detected by ELISA (Fig. 5c, d). When Pfs230-D1 and Pfs48/45-D3 were combined as cocktails or mosaics on ferritin or I53-50, ELISA signals remained consistent with those of the respective single-antigen immunogens. Notably, for STAC presented on *H. pylori* ferritin or I53-50, Pfs230-D1 titers remained comparable to those of single-antigen immunogens, whereas sera reactivity to Pfs48/45-D3 was lower (Fig. 5c, d). To further characterize the immunodominance hierarchies of epitopes on STAC, we depleted mouse sera after immunization with STAC on *H. pylori* ferritin using monomeric STAC, Pfs230-D1, Pfs48/45-D3, or a combination of Pfs230-D1 and Pfs48/45-D3, followed by ELISA to detect remaining STAC-specific antibodies. Depletions with STAC,

Pfs230-D1, or the combination of Pfs230-D1 and Pfs48/45-D3 removed >99%, 97%, and 99% of STAC-specific antibodies, respectively, while depletion with Pfs48/45-D3 alone only depleted 44% of the STAC-reactive antibodies, similar to the 35% depleted after the negative-depletion control (Sup Fig. 10c). TRA was evaluated using the depleted sera from each depletion group by SMFA. Total sera, Pfs48/45-depleted, or negative-depleted control exhibited TRA values of 98.3%, 94.2%, and 93.2%, respectively (Sup fig 10d). In contrast, no detectable TRA was measured for sera depleted for STAC-, Pfs230-D1-, or both Pfs230-D1 and Pfs48/45-D3-specifc antibodies (Sup Fig.10d). Together, these data indicate that the mouse antibody response and corresponding inhibitory activity are predominantly directed to the Pfs230-D1 domain of STAC, consistent with results from STAC immunization on CPQ nanoparticles (Fig. 4).

Functional immunogenicity elicited by each immunogen group was evaluated by SMFA with serial dilutions of pooled sera. SMFAs performed with sera diluted 27-fold demonstrated >80% TRA across all antigen-containing I53-50 immunogen groups, whereas sera from negative controls exhibited no TRA (Fig. 5e). At 81-fold dilution, sera from the I53-50 immunogen group retained strong TRA, while the Pfs230-D1 and Pfs48/45-D3 groups both dropped below 80% TRA, and the mosaic and cocktail groups also saw modest decreases in activity. Notably, at 243-fold dilution, only the STAC group retained more than 80% TRA, and had significantly higher TRA than other groups (Fig. 5f). In contrast to the I53-50 immunogens, sera diluted 81-fold from all antigen-containing *H. pylori* ferritin immunogen groups demonstrated >80% TRA (Fig. 5g), potentially due to the engineering of N-linked glycans and helper T cell antigen in the *H. pylori* ferritin component. As seen with the I53-50 immunogens, at 243-fold dilutions of sera elicited by *H. pylori* ferritin immunogens, only the STAC group demonstrated >80% TRA, and this group had significantly higher TRA than the other groups (Fig. 5h). Taken together, these data indicate that when formulated as protein nanoparticle immunogens, STAC can elicit a more potent functional response at higher sera dilutions when compared to Pfs230-D1 and Pfs48/45-D3, both as individual antigens and in combination as cocktails or mosaics.

## Discussion

This study demonstrates the development of STAC, which integrates key potent epitopes from Pfs230-D1 and Pfs48/45-D3 on a single recombinant protein to enhance TBV efficacy. We leveraged insights gathered from previous antibody structure-function studies across both antigens to provide the molecular blueprints necessary to design an epitope presentation approach that also masked non-functional epitopes. Iterative structure-guided engineering improved the biochemical tractability, stability, and expression yield of STAC while preserving epitope fidelity for equivalent or enhanced functional immunogenicity. Furthermore, multimerized STAC using liposomal adjuvants or protein nanoparticles demonstrated potent transmission-reducing activity, in some instances drastically outperforming individual antigen-based vaccines. The success of STAC in eliciting potent transmission-reducing antibodies highlights the potential for structure-guided vaccine optimization to improve antigen presentation and immune responses.

Clinical transmission-blocking vaccines (TBVs) that feature either Pfs230 or Pfs48/45 sub-domains outside the full protein context may suffer from eliciting immune responses towards non-functional epitopes. The Pfs230D1M-EPA candidate has been under evaluation in malaria-endemic areas using a 4-dose schedule and has been well tolerated, with sera samples also being collected for laboratory analysis (NCT05135273, NCT03917654). In the first study to examine human antibody responses to Pfs230-D1 vaccination, only 1 of the 9 identified Pfs230-D1-directed mAbs possessed appreciable TRA[26]. A recent antibody repertoire analysis derived from sequencing Pfs230D1-specific B cells in response to Pfs230D1M-EPA/Alhydrogel® and Pfs230D1M-EPA/

AS01 formulations days after dose 4 or dose 3, respectively, found the majority of the 65 Pfs230D1-directed mAbs lacked functional activity when assessed by SMFA using a cutoff of 75% TRA at 100 μg/ml of purified antibody[23]. Multiple nonfunctional mAbs, including LMIV230-02, utilize the IGHV1-69 germline sequence and share an epitope bin with LMIV230-02[33], which we have structurally mapped to a buried C-terminal epitope exposed in the Pfs230-D1 subunit vaccine. While Pfs230-D1 is an important antigen for TBV development, these findings from the Pfs230D1M-EPA trials identify a critical opportunity for epitope-masking strategies like STAC to overcome this limitation of domain-based subunit vaccines.

Ongoing phase I trials for Pfs48/45 candidates provide opportunities to elucidate the human antibody repertoires against this subunit vaccine antigen. At present, a comprehensive epitope map of Pfs48/45-D3 is not available, with multiple surfaces unmapped for any known antibody epitope bins. Structural elucidation of full-length Pfs48/45 in complex with Pfs230 suggests certain epitopes on Pfs48/45-D3 may be occluded by Pfs230 in the heterodimer complex[18,19]. These would become exposed in a Pfs48/45 domain-based subunit vaccine format, conceivably leading to the elicitation of non-functional antibodies directed to natively buried epitopes, as was the case for the Pfs230-D1 subunit vaccine. Moreover, given the limited clinical data available, it remains unclear whether Pfs230-D1 or Pfs48/45-D3 elicits the stronger functional immune response in malaria-endemic populations, where seroprevalence of antibodies to either Pfs230, Pfs48/45, or both antigens has been reported[29,30], underscores the challenge of selecting an optimal TBV candidate, and highlight the advantage of an approach that instead focuses on combining potent epitope bins present across both antigens.

Simultaneously targeting two antigens may enhance vaccine efficacy and durability as titers wane over time. Increased TRA has indeed been reported for antibody combinations targeting Pfs48/45 and Pfs230[36]. Furthermore, the candidate ProC6C, which fuses the Pfs230 Pro-domain to Pfs48/45-D3, induced superior TRA compared to R0.6C, with further enhancement upon multimerization as a virus-like particle (VLP)[39]. By combining the complement-dependent and independent epitopes of Pfs230-D1 and Pfs48/45-D3[32–38], respectively, two distinct TRA mechanisms are simultaneously engaged, potentially helping to circumvent immune evasion such as potential genetic drift and complement escape[57–59]. In addition, a single chimeric protein, instead of two separate immunogens, is likely to reduce the burden of production, which may lead to a lower cost of goods.

The next-generation TBV candidate described, STAC, further employs interdomain antigen scaffolding to mimic native protein conformations and reduce the exposure of non-native surfaces not known to be recognized by potent inhibitory antibodies. This strategy of immunofocusing away from undesired epitopes is functionally analogous to loop deletion, domain removal, and glycan repositioning −approaches that have been shown to enhance humoral responses in viral vaccine antigens, including influenza virus[60], HIV-1[61–65], SARS-CoV-2[66,67], and RSV[68,69]. By mitigating non-functional antibody interference, STAC may overcome limitations of leading TBV candidates and represent an important epitope-masking strategy with broader applications in vaccine development. Importantly, the genetic conjugation of distinct epitopes may confer immunodominance hierarchies between epitopes, as we have observed in rodent immunizations with STAC, which elicits a Pfs230-D1-dominant response. Several factors may contribute to this phenomenon, including germline-encoded bias of specific B cell lineages for residue signatures at specific epitopes[70], proteolysis sensitivity in B cell follicles for specific epitopes[71] or epitope prominence by antigen orientations on nanoparticles[72]. Investigating the rules of immunodominance for different epitopes, and their contributions to function across preclinical species and in the context of the human Ig repertoire, will be important next steps to further understand responses to STAC-based immunogens, and whether

further engineering is warranted to obtain a more balanced response to the potent Pfs230 and Pfs48/45 epitopes in a human Ig repertoire context.

In conclusion, this study provides a compelling case for structure-guided antigen engineering in malaria TBV development. By integrating an epitope-focused antigen presentation with multimerization strategies, STAC emerges as a promising candidate for improving transmission-blocking efficacy. The high potency of STAC as a protein nanoparticle immunogen also opens avenues for RNA-based delivery[73], which has shown promise for TBVs[74,75] and other malaria vaccine candidates[76–78]. Finally, evaluating STAC alongside pre-erythrocytic and blood-stage antigens, such as PfCSP and PfRH5, has the potential to inform the development of multistage malaria vaccines. Given the persistent challenge of malaria transmission in endemic regions, incorporating STAC into broader control strategies could represent a pivotal step toward sustainable malaria elimination.

## Methods

### Recombinant protein expression and purification

**Cell lines.** For expression of recombinant proteins, female human cell lines (HEK 293 F, FreeStyle™ 293-F cells, Thermo Fisher Scientific; HEK 293S, GnT I$^{-/-}$ cells, ATCC) were cultured in suspension in GIBCO™ FreeStyle™ 293 Expression Medium (Thermo Fisher Scientific) for 6–7 days at 37 °C, with 70% humidity and 8% $CO_2$ and rotating at 150 rpm. These cell lines were not authenticated beyond the manufacturer.

**Antigen monomers.** All Pf antigens were based on the consensus 3D7 sequence, gene synthesized and cloned into the pcDNA3.4 expression vector (GeneArt) and expressed and purified as previously described[37]. Briefly, antigens were transiently expressed in HEK293F or HEK293S cells (ThermoFisher Scientific) and purified using a 5 mL HisTrap FF column (GE Healthcare), followed by size exclusion chromatography (Superdex 200 Increase 10/300 GL, GE Healthcare). Sequences of designed antigens are available in Supplementary Fig. 6.

**Antibodies.** Antibodies were expressed and purified as explained previously[37]. Briefly, variable light (VL) and heavy (VH) chains of Fabs used in these studies were gene-synthesized and cloned (GeneArt) into custom pcDNA3.4 expression vectors immediately upstream of human Igκ and Igγ1-CH1 domains. Fab heavy chain and Fab light chain plasmids were co-transfected at a 2:1 ratio into FreeStyle 293-F cells (Thermo Fisher Scientific) at a cell density of $0.8 \times 10^6$ cells/ml for transient expression using PEI MAX transfection reagent (Polysciences). Cells were cultured in GIBCO FreeStyle 293 Expression Medium for 6–7 days, and supernatants were isolated by centrifugation and filtered through a 0.22-μm membrane. Fabs were purified following a scheme of HiTrap KappaSelect affinity chromatography (Cytiva) and cation exchange chromatography (MonoS, Cytiva).

***H. pylori* ferritin protein nanoparticles.** Antigen-conjugated nanoparticles containing the mutations from native *Helicobacter pylori* ferritin sequence (K75N, E77T, E99N, and I101T) to add two N-linked glycan sequons, as well as addition of a PADRE sequence at the C-terminus (AKFVAAWTLKAAA) were transiently expressed in HEK293F cells (ThermoFisher Scientific) in the presence of 10 μM Kifunensine (Cayman Chem) to obtain high-mannose glycans. All purification instruments were washed with 500 mM sodium hydroxide and purified water prior to protein isolation to avoid endotoxin contamination. Purification was completed using a 5 mL HisTrap FF column (GE Healthcare), followed by size exclusion chromatography (Superose 6 10/300 GL, GE Healthcare). Endotoxin was assessed using the EndoSafe Nexgen-PTS System (Charles River). The threshold for samples suitable for immunization was <5 EU/ml. If required, endotoxin removal using a ToxinEraser endotoxin removal kit (GenScript) was performed, and endotoxin levels below threshold were

reconfirmed. Antigen-conjugated nanoparticle structural integrity was confirmed by negative stain electron microscopy. Proteins were diluted to 20–50 μg/ml and adsorbed onto homemade carbon film-coated grids, which were stained with 2% uranyl formate. Specimens were imaged with a Hitachi HT7800 electron microscope operating at 120 kV with a Xarosa CMOS camera.

**I53-50 protein nanoparticles.** Plasmids encoding antigen-conjugated I53-50A and I53-50B.4PT1 were synthesized by GenScript and cloned into pET29b between the NdeI and XhoI restriction sites. I53-50B.4PT1 incorporated a double stop codon directly before the C-terminal polyhistidine tag to make a tagless construct. Antigen-conjugated I53-50A proteins were produced in Expi293F cells grown in suspension using Expi293F expression medium (Life Technologies) at 33 °C, 70% humidity, 8% $CO_2$, rotating at 150 rpm. Cultures were transfected using PEI-MAX (Polyscience) with cells grown to a density of 3.0 million cells per mL and cultivated for 3 days. Supernatants were clarified by centrifugation (5 min at 4000 x $g$), the addition of PDADMAC solution to a final concentration of 0.0375% (Sigma Aldrich, #409014), and a second spin (5 min at 4000 x $g$). Proteins containing His tags were purified from clarified supernatants via an AKTA AVANT FPLC using prepacked 5 mL Ni Sepharose Excel resin columns (Cytiva), with each clarified supernatant supplemented with 1 M Tris-HCl, pH 8.0, to a final concentration of 45 mM and 5 M NaCl to a final concentration of ~310 mM, prior to column application. The resin was washed with 5 column volumes (CV) of 20 mM Tris pH 7.0, 150 mM NaCl, and the protein was eluted over 10 CV up to 100%B buffer, which was composed of 20 mM Tris pH 7.0, 150 mM NaCl, 500 mM imidazole. The pentamer was expressed in Lemo21(DE3) cells (NEB) in LB medium (10 g Tryptone, 5 g Yeast Extract, 10 g NaCl) in a 10 L BioFlo 320 Fermenter (Eppendorf). At inoculation, culture conditions were set to 37 °C, 225 RPM impeller speed, and 5 SLPM (standard liter per minute) gas flow, with $O_2$ supplementation active as part of the dissolved oxygen cascade. At the onset of a dissolved oxygen spike (OD ~ 12), the culture was supplemented with 100 mL of 100% glycerol and induced with 1 mM IPTG. Simultaneously, the temperature was lowered to 18 °C, and $O_2$ supplementation was stopped. Protein expression continued overnight until reaching an OD ~ 20. Cells were harvested by centrifugation, and pellets were resuspended in dPBS (Cat# 14190144), homogenized, and lysed using a Microfluidics M110P microfluidizer at 18,000 psi for three passes. Lysates were clarified by centrifugation (24,000 × $g$, 30 min, 4 °C), and the supernatant was discarded to isolate inclusion bodies. Inclusion bodies were first washed with dPBS, 0.1% Triton X-100 (pH 8.0), followed by centrifugation for sample clarification. The resulting pellet was then washed with dPBS supplemented with 1 M NaCl (pH 8.0). After the second wash, pentamer protein was extracted from the pellet using dPBS, 2 M urea, 0.75% CHAPS (3-[(3-Cholamidopropyl) dimethylammonio]-1-propanesulfonate), pH 8.0. Extracted protein was loaded onto a DEAE Sepharose FF column (Cytiva) using an AKTA Avant150 FPLC system (Cytiva). After binding, the column was washed with 5 CV of dPBS + 0.1% Triton X-100 (pH 8.0), followed by 5 CV of dPBS + 0.75% CHAPS (pH 8.0). Elution was performed using 3 CV of dPBS + 500 mM NaCl (pH 8.0). Eluted fractions for both antigen-conjugated I53-50A and I53-50B.4PT1 were confirmed by SDS-PAGE, pooled, concentrated using 10 K MWCO centrifugal filters (Millipore), sterile-filtered (0.22 μm), and applied to a HiLoad S200 pg GL SEC column (Cytiva) using either dPBS or 50 mM Tris pH 8, 500 mM NaCl, 0.75% CHAPS buffer, respectively. After sizing, peak fractions for each protein were confirmed by SDS-PAGE, pooled, concentrated using 10 K MWCO centrifugal filters (Millipore) to 50 μM, sterile-filtered (0.22 μm), and assessed for low endotoxin content before freezing in single-use aliquots stored at −80 °C. Endotoxin levels in protein samples were measured using the EndoSafe Nexgen-MCS System (Charles River). Samples were diluted 1:50 in Endotoxin-free LAL reagent water and applied to wells of an EndoSafe LAL reagent cartridge. Charles

River EndoScan-V software was used to analyze endotoxin content, automatically back-calculating for the dilution factor. Endotoxin values were reported as EU/mL, which were then converted to EU/mg based on UV/vis measurements. The threshold for samples suitable for immunization was <10 EU/ml.

**Liposome nanoparticles.** CPQ liposomes were prepared as previously described[46] with 1,2-Dioleoyl-sn-glycero-3-phosphocholine (DOPC; Corden, catalog number LP-R4-070), cholesterol (CHOL; PhytoChol, Evonik), Monophosphoryl Hexa-acyl Lipid A, 3-Deacyl (PHAD-3D6A; Avanti catalog number 699855), and QS-21 (Desert King). Liposomes had a [DOPC: CHOL: CoPoP: PHAD-3D6A:QS-21] mass ratio of [20:5:1:0.4: 0.4] and were prepared using ethanol injection and nitrogen pressurized extrusion through stacked 200 nm, 100 nm, and 80 nm polycarbonate membranes (Whatman) in PBS, carried out at 50 °C. Ethanol was then removed by dialysis against PBS twice at 4 °C. QS-21 (1 mg/mL) was added to the liposomes after formation. Liposomes were sterile filtered and stored at 4 °C. Liposome sizes were determined by dynamic light scattering (DLS) with a NanoBrook 90 plus PALS instrument after 200-fold dilution in PBS. To assess the stability of constructs of interest bound in particle form, a Ni-NTA competition assay was carried out as previously described[42]. Ni-NTA magnetic beads (ThermoFisher Scientific, catalog no. 88831) were added to the CPQ liposome-incubated antigen constructs (1:4 mass ratio of total protein:CoPoP) and incubated with the beads for 30 min at RT, then the supernatant and beads were separated using a magnetic separator (ThermoFisher Scientific, catalog no. 12321D). Denaturing reducing loading dye was then added to the samples (supernatant or beads) and heated at 95 °C for 10 min. The samples were then subjected to SDS-polyacrylamide gel electrophoresis (SDS-PAGE) using Novex 4 to 12% bis-tris acrylamide gels (Invitrogen, catalog no. NP0321BOX) and MES running buffer (Invitrogen, catalog no. NP0002).

### Animal immunizations
**Protein nanoparticles.** Female Crl:CD1(ICR) outbred mice were purchased from Charles River (order code 022) at 7 weeks of age and were housed in a specific pathogen-free facility within the Department of Comparative Medicine at the University of Washington, Seattle. Animals had access to standard rodent chow and water *ad libitum* and were maintained in an ABSL1 housing room with environmental parameters set to 68–79 °F, 30–70% humidity, and a dark/light cycle of 10 h of dark and 14 h of light. For each immunization, low-endotoxin immunogens were mixed with 1:1 vol/vol AddaVax (InvivoGen vac-adx-10) to reach the specified final dose for each immunogen per 0.1 mL injection. At 9 weeks of age, 5 mice per group were injected subcutaneously in the inguinal region at week 0 for the prime immunization and at 13 weeks of age for the boost immunization at week 4. Mice were bled via the submental route at weeks 0 and 2 and terminally at week 6. Blood was collected in serum separator tubes (BD # 365967) and rested for 30 min at room temperature for coagulation. Serum tubes were then centrifuged for 10 min at 2000 x *g*, and serum was collected and stored at −80 °C until use.

**Lipid nanoparticles.** The immunogenicity of Pfs230-D1, Pfs48/45-D3, and various designed antigen constructs was assessed using CPQ liposomes. To prepare the final vaccine, CPQ liposomes were incubated with the constructs for 3 h at RT. For each group, six 6–8 week old female CD-1 mice (Envigo) were immunized intramuscularly with indicated doses of protein bound to liposomes in a prime/boost dose regimen at day 0 and 21; sera were collected at day 42.

**Ethical Statement.** Animal experiments involving protein nanoparticles were conducted in accordance with the University of Washington's Institutional Animal Care and Use Committee (IACUC

protocol 4470-01). The University of Washington's animal use program is accredited by the Association for Assessment and Accreditation of Laboratory Animal Care (Reference Assurance: #000523). Immunization studies involving lipid nanoparticles were carried out in compliance with the University at Buffalo IACUC protocols (protocol BME05044Y).

### ELISA, antibody depletion and SMFA
**Lipid nanoparticles.** All ELISA were conducted using Pfs230-D1 or Pfs48/45-D3 as coating antigens using the method described previously[42]. For the ELISA data, log-transformed values were compared by a Student's *t*-test (two groups), or one-way ANOVA followed by Tukey's multiple comparison tests (more than two groups). The ability of antibodies to reduce the development of *P. falciparum* NF54 strain oocysts in the mosquito midgut was evaluated by SMFA as described elsewhere[48]. In brief, IgG was purified from whole sera using Protein G affinity chromatography, and then indicated concentrations of IgG were mixed with 0.15%–0.2% stage V gametocytemia and then fed to 3-6-day-old female *Anopheles stephensi*. All experiments were done in the presence of human complement. The mosquitoes were maintained for 8 days and then dissected to count the number of oocysts per midgut in 20 mosquitoes. For the statistical analysis of SMFA results, a zero-inflated negative binomial model was used[48].

**Protein nanoparticles.** To measure antibody titers in total sera, 50 µL of 2 µg/mL of Pfs230-D1 or Pfs48/45-D3 was plated onto 96-well Nunc Maxisorp (ThermoFisher) plates in TBS. Plates were incubated at 25 °C for 1 h, then blocked with 200 µL of 2% BSA in TBST for an additional 1 h at 25 °C. Plates were washed 3 × in TBST using a plate washer (BioTek), 1:5 serial dilutions of mouse sera were made in 50 µL TBST starting at 1:100 and incubated at 25 °C for 1 h. Plates were washed 3 × in TBST, then anti-mouse horseradish peroxidase-conjugated horse IgG (Cell Signaling Technology, #7076S) was diluted 1:2,000 in 2% BSA in TBST, and 50 µL was added to each well and incubated at 25 °C for 30 min. Plates were washed 3 × in TBST, and 100 µL of TMB (SeraCare) was added to every well for 2 min at room temperature. The reaction was quenched with the addition of 100 µL of 1 N HCl. Plates were immediately read at 450 nm on a SpectraMax M5 plate reader (Molecular Devices), and data were plotted and fit in Prism (GraphPad) using nonlinear four-parameter logistic sigmoidal regression, where X is log(concentration) to determine EC50 values from curve fits. For depletion experiments, NHS-activated magnetic beads (Pierce) were coated with Pfs230-D1, Pfs48/45-D3, STAC or PBS, following the manufacturer's instructions. After coupling, the beads were washed and stored in PBS. Sera were incubated with beads for 1.5 h at room temperature while gently mixing the suspension. After incubation, beads were separated on a magnetic stand, and serum was collected for analysis by ELISA and SMFA. One batch of serum was depleted with beads coated with Pfs48/45-D3, followed by another round of depletion with beads coated with Pfs230-D1. Depletion of antigen-specific antibodies was tested by ELISA. In short, 96-well Nunc Maxisorp (ThermoFisher) plates were coated with 1 µg/mL STAC in PBS overnight at 4 °C. Plates were washed with PBS and blocked with 5% milk in PBS for 1 h at 25 °C. After washing three times with PBS, serial dilutions of (depleted) sera in 1% milk in PBST were added in duplicate and incubated for three hours at 25 °C. Plates were washed with PBS, and 1:3000 diluted polyclonal rabbit anti-mouse HRP (P0260, Dako) in PBST was added, and plates were then incubated for 1 h at 25 °C. After washing, the plates were incubated with TMB, and reactions were stopped with $H_2SO_4$. Plates were read at 450 nm. EC50 values were calculated using nonlinear four-parameter logistic sigmoidal regression in Prism (GraphPad). SMFA experiments were conducted as previously described[79]. Blood meals containing cultured *P. falciparum* NF54 gametocytes mixed with mouse sera were fed to *Anopheles stephensi* mosquitoes (Nijmegen colony). Serum dilution factors indicate

the proportion of serum to the total blood meal volume. All SMFA experiments were conducted in the presence of active human complement. For each condition, 20 fully-fed mosquitoes were dissected, and oocysts were counted 6-8 days after feeding. Transmission-reducing activity was estimated using a mix-effects negative binomial regression model as previously described[80].

## Biolayer interferometry

**Kinetics.** Biolayer interferometry (Octet RED96, Sartorius) experiments at 25 °C were conducted to determine the binding kinetics of antibody-antigen interactions using recombinant His-tagged antigen proteins and Fabs. Antigen protein was diluted in kinetics buffer (PBS, pH 7.4, 0.01% (w/v) BSA, 0.002% (v/v) Tween-20) at 20 µg/ml and immobilized onto Ni-NTA (NTA) biosensors (FortéBio) to a threshold binding signal of 1.0 nm. Following a 60 s baseline measurement, biosensors were dipped into wells containing a twofold dilution series of Fab, from 500 nM to 15.6 nM, to monitor the association rate and then dipped back into kinetics buffer to monitor the dissociation rate. Kinetics data were analyzed using FortéBio's Data Analysis software 9.0, and kinetic curves were fitted to a 1:1 binding model using at least four concentrations.

**Quantitation.** For quantification experiments, RUPA-97 or TB31F Fabs were diluted in kinetics buffer to a concentration of 20 µg/ml and immobilized on an anti-human Fab CH1 biosensor (Sartorius). Pfs230-D1(D1+) and Pfs48/45-D3 (6C.mAgE1) were serially diluted in clarified and filtered HEK293F supernatant to a concentration range of 0–50 µg/mL, which was used to generate a standard curve from the association rate. For supernatants of all antigen designs quantitated, the initial association slopes were used for calculations, and all analyses were done using Sartorius' Data Analysis software 9.0. To contrast the relative signals for each antigenic domain, relative binding signals were additionally calculated by normalizing each binding signal relative to the maximum observed binding signal for each specific antigen.

## X-ray crystallography

For the ternary complex of RUPA-39 and RUPA-44 Fabs bound to Design-4, a 1.5:1.5:1 molar ratio of Fab:Fab:antigen was used. RUPA-44 and Design-4 were both expressed from HEK293S cells and previously treated with EndoH to truncate N-linked glycans present on both molecules. For the complex of RUPA-39 Fab bound to Pfs230-D1+, a 1:1.5 molar ratio of Fab:antigen was used. Both Fab-antigen complexes were incubated for 1 h at 4 °C followed by size exclusion chromatography (Superdex 200 Increase 10/300 GL, Cytiva) in 20 mM Tris and 150 mM sodium chloride at pH 8.0. Purified RUPA-39:Pfs230-D1+ complex was concentrated to 9.2 mg/ml and mixed in a 1:1 ratio with crystallization buffer (0.2 M calcium acetate and 20 % (w/v) PEG 3350). Crystals grew from a sitting-drop by vapor diffusion at 20 °C and were cryoprotected in crystallization buffer supplemented with 15% ethylene glycol, before being flash-frozen in liquid nitrogen. Diffraction data were collected at the CMCF-ID beamline at the Canadian Light Source and then processed and scaled using XDS and Aimless. Purified RUPA-39:RUPA-44:Design-4 complex was concentrated to 13.0 mg/ml and mixed in a 1:1 ratio with crystallization buffer (0.1 M sodium cacodylate, pH 6.5, 40% PEG 300, 0.2 M calcium acetate). Crystals grew from a sitting-drop by vapor diffusion at 20ºC and were cryoprotected in crystallization buffer supplemented with 10% ethylene glycol, before being flash-frozen in liquid nitrogen. Diffraction data were collected at the 23-ID-D beamline at the Argonne National Laboratory Advanced Photon Source. The diffraction data was processed with XDS and scaled with correction for anisotropy using the Staraniso server to a diffraction-limit cutoff ($I_{mean}/\sigma(I_{mean})$) of 1.2. Both structures were determined by molecular replacement using Phaser[81]. Structural refinements were performed using PHENIX[82], and models were manually checked and improved with Coot[83]. Images were generated using PyMOL (The PyMOL Molecular Graphics System, v2.3.4, Schrödinger, LLC.). Access to all software was supported through SBGrid[84].

## Computational design

Mutable residue selection for the Design-4 interdomain interface was defined as residues containing sidechain atoms within 7 Å of the neighboring domain as observed from the crystal structure of Design-4 bound by RUPA-39 and RUPA-44 Fabs. For fixed-backbone sequence redesign, the crystal structure was used as input to ProteinMPNN (v1.0.1)[44] and the 34 positions located around the interdomain interface were selected for redesign. The input model backbone coordinates were either noised by 0.02 Å or left unaltered, and three temperatures (0.1, 0.2, and 0.3) were sampled using the default weights for all twenty amino acids, with 20 sequences generated per temperature for a total of 120 sequences generated using the v_48_020 model of ProteinMPNN. Generated sequences were ranked by the ProteinMPNN sequence likelihood score with preference for higher sequence recovery to ensure proper domain folding and down-selected to 15 sequences, which were predicted with AlphaFold2[85,86] using 6 recycling steps without structural templating. All 15 designs adopted domain folds that matched Pfs230-D1 and Pfs48/45-D3 crystal structures with Cα RMSD < 1.0 Å for beta-sheet residues and possess interface residues with pLDDT > 80, although interdomain orientations were not identical with the input structure for all designs. All 15 designed sequences were selected for expression testing.

## Cryo-EM

**Data collection.** The complex of the stabilized tandem antigen chimera (STAC) of Pfs230 and Pfs48/45 bound to RUPA-97, LMIV230-01, TB31F, and RUPA-44 was concentrated to 5 mg/ml with 0.13 CMC of DDM in 20 mM Tris and 150 mM sodium chloride at pH 8.0. 3.2 µl of the sample was deposited on glow-discharged homemade holey gold grids[87]. A Leica EM GP2 Automatic Plunge freezer operated at 4 °C and 90% humidity was used to blot away excess sample (2.7 s), and grids were subsequently plunge-frozen in liquid ethane. Data were collected on a Thermo Fisher Scientific Titan Krios G3 operated at 300 kV and equipped with a Selectris X energy filter with a slit width of 10 eV and a Falcon 4i camera. Data acquisition was automated with EPU software. A nominal magnification of 130,000× with a calibrated pixel size of 0.93 Å was used for data acquisition, and exposures were collected as 6.4 or 6.5 s movies with a total exposure of ~50 or ~53.7 electrons/Å². Movies were recorded in Electron Event Representation mode[88] and fractionated into 40 frames prior to data analysis. 4676 and 5995 raw movies were obtained at 0° and 35° tilt, respectively, across two data collections.

**Image processing.** Cryo-EM data were processed using CryoSPARC v4.5.2[89]. The movies from the un-tilted and 35° tilted datasets were corrected using patch motion correction, and contrast transfer function (CTF) parameters were determined using patch CTF estimation. A subset of un-tilted micrographs was used to pick and extract particles for 2D classification. Selected 2D classes were used as templates for the template picker on a set of 4085 un-tilted micrographs (CTF < 5 Å). 2X Fourier cropped particles were extracted, underwent 2D classification and selection, and were used for ab initio reconstruction. After heterogeneous refinement, particles from the selected class were re-extracted from micrographs without Fourier cropping (1,094,268 particles) and used for ab initio reconstruction with two classes, followed by heterogeneous refinement and non-uniform refinement to generate a 3.2 Å map (399,207 particles). This volume was used to generate templates for template particle picking on a curated dataset including both tilted and un-tilted data using thresholds for average defocus values (1000-30000 Å), CTF fit resolution (<3.5 Å for un-tilted micrographs, <4.2 Å for tilted micrographs), and relative ice thickness (0.95-1.13). A total of 5287 micrographs were accepted, including 3122

untilted and 2165 35° tilted micrographs. Template picker resulted in 2,674,017 extracted particles (360 × 360 px box size). 2D classification and selection were used to remove poor-quality particles, and the resulting particles were used for ab initio 3D reconstruction (done twice with two different sets of 2D classes selected). Generated maps were then used in several rounds of heterogeneous refinement to further classify particles. 2D classes corresponding to rare views of the complex were selected for Topaz[90] particle picking, training and extraction. After subsequent extraction and 2D classification, selected particles were combined with a previous set of 814,237 particles, followed by duplicate particle removal and heterogeneous refinement. Topaz training, particle picking, and heterogeneous refinement were done twice in parallel. Particles from selected heterogeneous refinement classes from each job were combined, and duplicates were removed, resulting in 2,216,915 particles. This set of particles was used for heterogeneous refinement with 5 volumes, and particles from one class (645,647 particles) were subsequently used for 3D classification with 5 classes. The selected class underwent one round of heterogeneous refinement, and the class corresponding to the complex was put through duplicate removal (147,546 particles), followed by non-uniform refinement. Local refinement with a mask containing the stabilized tandem antigen chimera and the variable regions of RUPA-97, LMIV230-01, TB31F, and RUPA-44 was done to achieve a 3.22 Å map (147,546 particles). A detailed workflow for cryo-EM data processing is depicted in Supplementary Fig. 8.

**Modeling building.** Starting structural models were obtained by fitting previously determined structural models (PDB: 6E63, Pfs48/45-D3:TB31F; 7UXL, Pfs48/45-D3:RUPA-44; 7UVQ, Pfs230-D1:RUPA-97; and 7UFW, Pfs230-D1D2:LMIV230-01) into the experimentally determined map. Model building and refinements were done using PHENIX[82] and Coot[83]. Images were generated using ChimeraX[91], and access to all software was supported through SBGrid[84].

**Small-angle X-ray scattering**
An Anton Paar SAXSpace instrument (Anton-Paar GmbH, Austria) with SAXSDrive software was used to collect SAXS scattering curves for monomeric antigen proteins at concentrations between 3–4 mg/ml in a 1x PBS buffer. SAXS data were collected in 6 × 10 min exposure frames at 20 °C under vacuum. The zero position for each scattering curve was calibrated using SAXSTreat before buffer subtraction using SAXSquant. SAXStreat software (Anton-Paar GmbH, Austria) was used to define the zero position of the scattering curve and convert the data to 1D scattering profiles. SAXSQuant (Anton-Paar GmbH, Austria) was then used to desmear the data and perform buffer subtraction. Analyses of the processed SAXS data were done using the ATSAS package (version 3.2.1; EMBL, Hamburg, Germany). PRIMUS[92] was used to perform Guinier analysis, and GNOM[93] was used to calculate the P(r) distance distribution. Additional SAXS data collection parameters are available in Supplementary table 4.

**Differential scanning calorimetry**
Antigen samples at 1 mg/ml were buffer exchanged into 1X PBS, and all measurements were conducted on a Nano DSC (TA Instruments) in duplicate, unless otherwise stated. The sample was heated from 20 to 95 °C at a constant heating rate of 1.0 °C/min. Runs with only 1x PBS buffer were performed and subtracted from the melting curves, and the data were fit to a single-stage unfolding model. The melting temperature ($T_m$) for each protein was determined by the NanoAnalyze software (TA Instruments) by fitting the data to the Gaussian curve with a melting temperature onset ($T_m^{onset}$) model.

**Reporting summary**
Further information on research design is available in the Nature Portfolio Reporting Summary linked to this article.

## Data availability
The biomolecular structural data generated in this study have been deposited in the RCSB Protein Data Bank under accession codes 9N8N, 9N8I, and 9N8J, and are publicly available as of the date of publication. The cryo-EM volume data has been deposited in the Electron Microscopy Data Bank under entry ID EMD-49130. This paper does not report original code. Raw SMFA data is available in the source data. Any additional information required to reanalyze the data reported in this paper is available from the corresponding author upon request. Source data are provided with this paper.

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

## Acknowledgments

We would like to thank Nicholas Proellochs, Wouter Graumans, Marga van de Vegte-Bolmer, Laura Pelser-Posthumus, Astrid Pouwelsen, Jacqueline Kuhnen and Jolanda Klaassen for assistance with parasite culture and mosquito infections; Samir Benlekbir and Zhijie Li from the SickKids Nanoscale Biomedical Imaging Facility for assistance and insights during cryo-EM data collection; Greg Wasney and James Magnus Jorgensen at the Structural & Biophysical Core (SBC) Facility for assistance. X-ray diffraction experiments for biomolecular crystallography were performed at GM/CA@APS, which has been funded in whole or in part with federal funds from the National Cancer Institute (ACB-12002) and the National Institute of General Medical Sciences (AGM-12006). The Eiger 16 M detector was funded by an NIH–Office of Research Infrastructure Programs High-End Instrumentation grant (1S10OD012289-01A1). This research used resources of the Advanced Photon Source, a U.S. Department of Energy (DOE) Office of Science user facility operated for the DOE Office of Science by Argonne National Laboratory under contract DE-AC02-06CH11357. X-ray diffraction experiments were also performed at beamline CMCF-ID at the Canadian Light Source, a national research facility of the University of Saskatchewan, which is supported by the Canada Foundation for Innovation (CFI), the Natural Sciences and Engineering Research Council (NSERC), the National Research Council (NRC), the Canadian Institutes of Health Research (CIHR), the Government of Saskatchewan, and the University of Saskatchewan. The small-angle X-ray scattering instrument was accessed at the Structural and

Biophysical Core Facility, The Hospital for Sick Children, and EM data was collected at the Nanoscale Biomedical Imaging Facility, The Hospital for Sick Children, supported by the Canada Foundation for Innovation and Ontario Research Fund. This work was supported by a National Institutes of Health grant (1R01AI148557-01A1 to J.F.L., R.S.M., and J-P.J.); Bill & Melinda Gates Foundation grant (OPP1156262 to N.P.K. and J-P.J.); a Canadian Institutes of Health Research Project grant (428410 to J-P.J.); and, in part, thanks to funding from the Canada Research Chair program (J-P.J.). M.M.J. is supported by the Netherlands Organization for Scientific Research (Vidi fellowship NWO project number 192.061). S.H. is supported by a Canada Graduate Scholarship - Doctoral. This work was also supported by the Division of Intramural Research (DIR), National Institute of Allergy and Infectious Diseases (NIAID).

## Author contributions

Experimental design was collaborative between all co-authors. Experiments were conducted by D.I., K.M., S.H., R.R., Y.S., W-C.H., R.S., K.T., G-J.v.G., E.M.L., S.C., C.M., A.S., and C.S. The manuscript was written by D.I. and J-P.J., and edited by all co-authors. Funding was secured by R.S.M., C.A.L., M.M.J., N.P.K., J.F.L., and J-P.J.

## Competing interests

The authors declare no competing interests. Patent applications have been filed that relate to this work.

## Additional information

[1]Program in Molecular Medicine, The Hospital for Sick Children Research Institute, 686 Bay Street, Toronto, ON M5G 0A4, Canada. [2]Laboratory of Malaria and Vector Research, National Institute of Allergy and Infectious Diseases, National Institutes of Health, 12735 Twinbrook Parkway, Rockville, MD 20852, USA. [3]Vaccine Research Center, National Institute of Allergy and Infectious Diseases, National Institutes of Health, Bethesda, MD 20814, USA. [4]Department of Biochemistry, University of Toronto, 1 King's College Circle, Toronto, ON M5S 1A8, Canada. [5]Department of Biochemistry, University of Washington, Seattle, WA 98195, USA; Institute for Protein Design, University of Washington, Seattle, WA 98195, USA. [6]Department of Biomedical Engineering, University at Buffalo, State University of New York, Buffalo, NY 14260, USA. [7]Department of Medical Microbiology, Radboud University Medical Center, Nijmegen, the Netherlands. [8]Center for Vaccine Innovation and Access, PATH, 455 Massachusetts Avenue NW Suite 1000, Washington, DC 20001, USA. [9]Department of Immunology, University of Toronto, 1 King's College Circle, Toronto, ON M5S 1A8, Canada. [10]Centre for Global Child Health, The Hospital for Sick Children, Toronto, ON M5G 0A4, Canada. ✉e-mail: jean-philippe.julien@sickkids.ca

