## [Peer Review file · Nature Communications]

A stabilized tandem antigen chimera that elicits potent malaria transmission-reducing activity

Corresponding Author: Dr Jean-Philippe Julien

Version 0:

Reviewer comments:

Reviewer #1

(Remarks to the Author)

The manuscript by Ivanochko et al presents an interesting protein design story. Two related proteins, Pfs230 and Pfs48/45 each contain domains which elicit antibodies which can block transmission of malaria parasites and a structural comparison suggested to the authors that two of these domains can be combined into a novel chimeric antigen, while still presenting the key epitopes for transmission-blocking antibodies.

They conducted a stepwise design of this chimera, using a fusion, a linker design and finally a Protein MPNN-based design of the interface, validating at each step using mAb binding. They also used structural biology during the design process and at the end to validate the final design. This is a high-quality piece of work, with the design and validation well-presented and the data of high quality too. It resulted in a successful design.

The authors then tested their design ("STAC"), either attached to liposomes or fused to I53-50 or ferritin nanoparticles. On liposomes, STAC performed similarly to Pfs230D1. On ferritin it also behaved similarly, albeit with some statistically significant improvement at one concentration point. On I53-50, STAC seemed substantially improved over the individual components, generating transmission blocking activity which survived greater dilution.

This is well conducted study, with an interesting design and STAC shows potential for use in future transmission blocking vaccine strategies. I have one experiment which I would strongly recommend. The sera generated and tested in Figure 5 could be used to assess whether the transmission blocking activity induced by STAC is primarily due to antibodies targeting 230 or 48/45 or equally due to both. To do this, the authors could simply deplete antibodies from these sera against either 230D1 or 48/45D3 and assess the TRA of what is left. This would give an interesting answer about the relative importance of the two fusion partners in STAC. In my view, this would be a very interesting experiment which would complete this high-quality study.

Some questions:

In the first experiment, liposomes contained either the 230 antigen or the 48/45 antigen, or both mixed. Why did the authors not mix liposomes containing just 230 with liposomes containing just 48/45? As a B cell makes one type of antibody would it not be interesting to see if mixing two populations of liposomes, each liposome containing one antigen had a different effect to having all liposomes containing both antigens? Could the authors comment on why they didn't do this? Also the lower efficacy of the dual-conjugated over 230 alone seems to not justify the subsequent approach?

In the design the authors sensibly present that the key TB antibodies should still be able to bind. However, there might be antibodies not structurally characterised which could also be useful for transmission block. Are they likely to bind? In the case of 48/45 could the authors estimate what fraction of the surface area of 48/45 D3 exposed in 48/45 is also exposed in their tandem design?

Design-1 bound to the desired neutralising antibodies but also bound to the non-inhibitory antibodies, whose epitopes should have been blocked in the new tandem construct. The authors make it clear that this indicates that further design is needed (lines 246-8). It would be good to be more explicit here. I think that these means that the two domains are not associated in the correct relative orientation but instead are flexibly linked so that the epitopes are not occluded? Also 278.

Line 408-9 is too strongly worded. STAC only strongly outperforms on I53-50, with perhaps improvement (at one dilution point) for ferritin and none for liposomes. I think that the final conclusion should therefore be toned down a little.

Minor comments:

Figure 1c – could do with x axis labels for the black, yellow and blue circles.

Figure 1 – why not show binding data for the transmission blocking antibodies as well as the non-blockers which shouldn't bind?

Throughout, it would help the reader to have a clearer explanation of what the different mAbs do/bind. More "Pfs230-binding mAb XXXX" etc...

Figure 4b, should that be STAC not design-22?

Figure 4f is missing x axis labels for the proteins.

Reviewer #2

(Remarks to the Author)

This paper generates a chimeric molecule combining two leading transmission-blocking vaccine candidates, Pfs230D1 and Pfs48/45-D3. These subdomains have been previously described to contain critical protective epitopes and have been used to elicit antibodies that block gametocyte fertilization in mosquitoes in complement-dependent (Pfs230D1) and complement-independent fashion, Pfs48/45. These candidate antigens are expressed on late-stage gametocytes, are immunogenic, and can boost antibody responses to vaccines based on these domains. This study uses a detailed structure study to generate a chimeric molecule that optimizes functional epitopes while masking epitopes that elicit non-functional Abs. They have successfully and convincingly accomplished this goal. The structure study and analysis well be done and clever. Overall comments on the structure studies initially described and followed by comments on the immunological and functional aspects.

Comments related to the structural design

The authors of this study report a stabilized tandem antigen chimera of Pfs230-D1 (179 residues) and Pfs48/45-D3 (137 residues), representing the most potent epitopes of each antigen, thereby removing distracting epitopes from both. The individual constructs were combined with liposomes and used to immunize mice. They then examined the potential of linking both antigens in a single construct and evaluated, using BLI and known mAbs, the binding to the construct to confirm correctly folded domains and the masking of unwanted epitopes.

Design4 resulted in a crystal structure that is indistinguishable by rmsd from the native individual structures of Pfs230 and Pfs48/45, indicating correctly folded proteins despite linking them together.

In a next step, the authors applied AI-supported methods ProteinMPNN to improve the interface design and reduce unwanted binding, resulting in additional suitable designs. The improved designs increased expression yields and binding affinities to the tested mAbs. It would be interesting to note if the authors used the default weights for amino acids for their interface design or skewed towards residues typically found in interfaces.

Paragraph 304-307

How many residues are involved in the interface in total? Are the 16 introduced mutations 20% of the interface, or is it more like 80%? This shows how dynamic an interface could be, redesigned while maintaining or improving the interface interaction so that others can derive their own interface designs for other proteins of interest.

Also there is a discrepancy in Figure 3c showing 18 mutated residues instead of 16?

9N8I

Chain A and E might be worth revisiting Molprobity to examine those atom clashes and potentially choose a different rotamer to improve the structure if the electron density supports that conformation.

This seems to be supported by section 6.1 with the rotamer outliers as well for Chain E

Also Rfree/Rwork seem to suggest a slight overfit, might be better to cut the resolution limit to 1.9 Å instead of 1.85 Å

9N8N

Rotamers could be optimized.

Why did the authors model certain residues differently than the sequence construct suggests? I'm referring to the Table on Page 5, particularly residues 186, 195, 284, and 289.

Again it would be worth running Molprobity to optimize the rotamer outliers of Chain I

Minor

Line 175 define TRA

Comments related to immunology and the functional characteristics of the vaccine design

The standard membrane feeding assay (SMFA) is used to evaluate IgG isotypes enriched from mice after vaccination with various constructs to measure the reduction in oocyte formation. (Do you think other isotypes might participate for protection?) These assays are well executed and are the typical functional tests to assess transmission-blocking activity. There was one biological replicate. The best chimeric molecule, STAC, shows comparable transmission-reducing activity

(TRA) to the individual component antigens. In fact, its TRA is slightly lower than Pfs48/45 as a single molecule, suggesting that. There is only a slight improvement in TRA with STAC linked to nanoparticle immunization compared to the individual immunogens based on the SMFA model. Still, as the authors point out, the molecule is easy to produce and stable, which is important for vaccine candidate developability.

The other aspect of the study was a vaccine molecule designed to mask the stimulation of non-functional antibodies. Theoretically, this is important, as a non-functional antibody could cause a steric hindrance to functional antibodies. Is there data to support this concerning Pfs230 or Pfs48/45? If so, this should be discussed or highlighted as a limitation. For example if you add non-blocking mAbs to antibodies elicited by the different constructs do you observed alterations in TRA activity?

Minor points

1. In Fig. 2, the relative binding shown in c and h can be better described. Is this relative binding of the new construct to the Pfs230 or Pfs48/45 individually? Please make this clearer.
2. In some figures they used Design-22 and STAC interchangeably – this can be confusing. Please stick to one or the other designation as I presume these are the same constructs.
3. Figure 5 is critical. The different color codes are difficult to follow, particularly with small figures with poor resolution. Suggest using different shapes and colors for the symbols denoting the different constructs.

Reviewer #3

(Remarks to the Author)

Version 1:

Reviewer comments:

Reviewer #1

(Remarks to the Author)

The authors have largely responded positively to my review and have made changes to the manuscript.

The one area in which I don't agree with their response relates to the depletion experiments. The aim of their overall study is to generate an immunogen which combines the best parts of Pfs48/45 and Pfs230 into a single immunogen which elicits inhibitory antibodies against both components. I therefore suggested a depletion experiment in which they deplete sera for Pfs230D1 and Pfs48.45 antibodies and see the contribution to SMFA, assessing whether they have achieved their objectives of raising both sets of antibodies. They have done this experiment and present it in the response to reviewers, but prefer to save it for a future paper.

The data is nice and makes it clear that TRA is almost entirely due to Pfs230D1 antibodies. This shows that the strategy to design an immunogen which elicits transmission blocking antibodies against both Pfs230 and Pfs48/45 has not been entirely successful as transmission blocking antibodies have not been elicited against Pfs48/45.

In the end, this is an editorial decision. However, my view is that this data should be included and discussed in the paper as it directly relates to whether the aims of this study have been achieved.

Reviewer #2

(Remarks to the Author)

The authors have adequately address our concerns

Reviewer #3

(Remarks to the Author)

Reviewer #1 (Remarks to the Author):

The manuscript by Ivanochko et al presents an interesting protein design story. Two related proteins, Pfs230 and Pfs48/45 each contain domains which elicit antibodies which can block transmission of malaria parasites and a structural comparison suggested to the authors that two of these domains can be combined into a novel chimeric antigen, while still presenting the key epitopes for transmission-blocking antibodies.

They conducted a stepwise design of this chimera, using a fusion, a linker design and finally a Protein MPNN-based design of the interface, validating at each step using mAb binding. They also used structural biology during the design process and at the end to validate the final design. This is a high-quality piece of work, with the design and validation well-presented and the data of high quality too. It resulted in a successful design.

The authors then tested their design ("STAC"), either attached to liposomes or fused to I53-50 or ferritin nanoparticles. On liposomes, STAC performed similarly to Pfs230D1. On ferritin it also behaved similarly, albeit with some statistically significant improvement at one concentration point. On I53-50, STAC seemed substantially improved over the individual components, generating transmission blocking activity which survived greater dilution.

This is well conducted study, with an interesting design and STAC shows potential for use in future transmission blocking vaccine strategies. I have one experiment which I would strongly recommend. The sera generated and tested in Figure 5 could be used to assess whether the transmission blocking activity induced by STAC is primarily due to antibodies targeting 230 or 48/45 or equally due to both. To do this, the authors could simply deplete antibodies from these sera against either 230D1 or 48/45D3 and assess the TRA of what is left. This would give an interesting answer about the relative importance of the two fusion partners in STAC. In my view, this would be a very interesting experiment which would complete this high-quality study.

We thank the Reviewer for their positive assessment of our study where we present a stepwise design strategy to fuse distinct malarial transmission-blocking antigens together, as a proof-of-principle approach to occlude a non-functional epitope on Pfs230-D1. Indeed, while STAC performed similarly to Pfs230-D1 when presented on lipid nanoparticles, we found that STAC elicited significantly more potent functional activity at a high serum dilution when presented on either Hp ferritin or I53-50 nanoparticles, when compared to Pfs230-D1 or Pfs48/45-D3 on the same platforms. We agree with the Reviewer on the importance of understanding the relative contribution of each component of STAC to the overall antibody response and transmission-reducing activity. In the Result section related to the protein nanoparticle immunizations (lines 400–403), we note the following characterization of the Pfs230- and Pfs48/45-specific immune response:

Notably, for STAC presented on H. pylori ferritin or I53-50, Pfs230-D1 titers remained comparable to those of single-antigen immunogens, whereas sera reactivity to Pfs48/45-D3 was lower. This suggests the immune response in mice predominantly targeted the Pfs230-D1 domain of STAC (Fig 5c), consistent with results from STAC immunization on CPQ nanoparticles (Fig 4).

We would like to thank the Reviewer for their suggestion of depletion assays to further interrogate the functional antibody responses presented in Figure 5. In fact, this is an experiment that has already been performed in the context of a follow-on study currently under way. Indeed, while the current manuscript aims to present the design of STAC and validate a non-inferior/superior immunogenicity and functional activity in WT mice, we had generated this depletion data as a component of a subsequent study dissecting the immunogenicity and individual contribution of each domain in mouse, rat, aotus and human Ig repertoire mouse at both the polyclonal and monoclonal levels. Given the ongoing nature of these experiments, and their extensive scope to properly characterize the immunodominance hierarchies of epitopes on STAC across preclinical species, we believe that this content is beyond the scope of the present article. Nonetheless, in transparency to the Reviewer, we are sharing the results of a depletion experiment from WT mouse sera after immunization with STAC on *H. pylori* ferritin, which further corroborates the reactogenicity data already present in the manuscript that the transmission-reducing activity elicited by STAC in WT mice is principally orchestrated by the Pfs230-D1-directed antibodies.

Figure showing reactogenicity (left) and functional activity (right) of WT mouse sera from immunization with STAC on *H. pylori* ferritin depleted with specific antigens. (Left) ELISAs demonstrating residual STAC-reactive sera after depletion with indicated antigens. Percentages indicate reduction in titers compared to those in total serum. (Right) TRA of the same depleted sera at an 81-fold dilution. TRA values are estimates from two independent experiments.

In the revised manuscript, we have added to the discussion of immunodominance in the Discussion section, as well as provided further clarity on the importance for future work in this area to better understand its implication (lines 480–490):

Importantly, the genetic conjugation of distinct epitopes may confer immunodominance hierarchies between epitopes, as we have observed here in WT mice immunizations with STAC that elicited a Pfs230-D1-dominant response. Several factors may contribute to this phenomenon, including germline-encoded bias of specific B cell lineages for residue signatures at specific epitopes⁷⁰, proteolysis sensitivity in B cell follicles for specific epitopes⁷¹ or epitope prominence by antigen orientations on nanoparticles⁷². Investigating the rules of immunodominance for different epitopes, and their contributions to function across preclinical species and in the context of the human Ig repertoire will be important next steps to further understand responses to STAC-based immunogens, and whether further engineering is warranted to obtain a more balanced response to the potent Pfs230 and Pfs48/45 epitopes in a human Ig repertoire context.

Some questions:

In the first experiment, liposomes contained either the 230 antigen or the 48/45 antigen, or both mixed. Why did the authors not mix liposomes containing just 230 with liposomes containing just 48/45? As a B cell makes one type of antibody would it not be interesting to see if mixing two populations of liposomes, each liposome containing one antigen had a different effect to having all liposomes containing both antigens? Could the authors comment on why they didn't do this? Also the lower efficacy of the dual-conjugated over 230 alone seems to not justify the subsequent approach?

To provide additional clarity as to the overall rationale of the study, i.e. including two antigens in one construct, we added to the Introduction (lines 131–135):

In addition, seroprevalence of antibodies to Pfs230 and Pfs48/45 indicate that some individuals in endemic regions elicit detectable antibody titers exclusively to one individual antigen but not to the other^{29,30}, so it is conceivable that an immunogen presenting both Pfs230 and Pfs48/45 could have the advantage of boosting the naturally acquired immunity for a broader diversity of individuals.

And to the Discussion (lines 457–462 and 470–472) with the following sentence:

Moreover, given the limited clinical data available, it remains unclear whether Pfs230-D1 or Pfs48/45-D3 elicits the stronger functional immune response in malaria-endemic populations, where seroprevalence of antibodies to either Pfs230, Pfs48/45, or both antigens has been reported^{29,30}. This underscores the challenge of selecting an optimal TBV candidate, and

highlight the advantage of an approach that instead focuses on combining potent epitope bins present across both antigens. [...] In addition, a single chimeric protein, instead of two separate immunogens, is likely to reduce the burden of production, which may lead to a lower the cost of goods.

By carrying out the first experiment with equivalent mass amounts of Pfs230-D1 and Pfs48/45-D3 on the same lipid nanoparticles, we did observe lower overall efficacy, which we rationalized was due to a sub-stoichiometric ratio of 0.74-to-1 of the larger antigen, Pfs230-D1. The experiment suggested by the Reviewer to compare a cocktail of two separate nanoparticle immunogens vs one nanoparticle containing two immunogens wasn't performed on the CoPoP platform, but was carried out in the context of the I53-50 nanocage (Fig. 5). As suggested by the Reviewer, these differential display approaches could lead to differences in the B cell response elicited, and we have expanded on this topic in the Discussion (lines 481–486):

Importantly, the genetic conjugation of distinct epitopes may confer immunodominance hierarchies between epitopes, as we have observed in rodent immunizations with STAC which elicits a Pfs230-D1-dominant response. Several factors may contribute to this phenomenon, including germline-encoded bias of specific B cell lineages for residue signatures at specific epitopes⁷⁰, proteolysis sensitivity in B cell follicles for specific epitopes⁷¹ or epitope prominence by antigen orientations on nanoparticles⁷².

In the design the authors sensibly present that the key TB antibodies should still be able to bind. However, there might be antibodies not structurally characterised which could also be useful for transmission block. Are they likely to bind? In the case of 48/45 could the authors estimate what fraction of the surface area of 48/45 D3 exposed in 48/45 is also exposed in their tandem design?

It is certainly possible that yet-undiscovered epitopes exist on Pfs48/45-D3, however functional antibodies are unlikely to bind to Pfs48/45-D3 at the interface with Pfs48/45-D2 due to steric occlusion – analogous to the non-functional antibodies which bind to the Pfs230-D1-D2 interface (e.g 15C5 and LMIV230-02). This is because the same surface on Pfs48/45-D3 that is occluded by Pfs230-D1 on STAC is also occluded by Pfs48/45-D2 in the native protein, thereby indicating a similar relative surface area exposure of Pfs48/45-D3 in both the native Pfs48/45 and STAC proteins. To address this important question raised by the Reviewer, we have incorporated a structural alignment of STAC with a crystal structure of full-length Pfs48/45 (PDB 7ZXF) at Supplementary figure 6f and updated the accompanying Results text (lines 318–321) with the following panel and sentence:

Notably, STAC exhibited distinct interdomain orientations when compared to crystal structures of Pfs230-D1D2 or Pfs4845-D1D2D3, although in both cases similar interdomain regions were occluded between either Pfs230-D1 or Pfs48/45-D3 and the adjacent domain (Sup fig 6e-f).

Design-1 bound to the desired neutralising antibodies but also bound to the non-inhibitory antibodies, whose epitopes should have been blocked in the new tandem construct. The authors make it clear that this indicates that further design is needed (lines 246-8). It would be good to be more explicit here. I think that these means that the two domains are not associated in the correct relative orientation but instead are flexibly linked so that the epitopes are not occluded? Also 278.

We fully agree with the Reviewer's description, and have integrated additional language in this revised Results section (lines 254–257):

In Design-1, we utilized the inter-domain linker sequence of Pfs230-D1D2 to connect Pfs230-D1 to Pfs48/45-D3, however partial accessibility of the Pfs230-directed 15C5 and LIMV230-02 epitopes indicated that flexible linkage may have mispositioned the two tandem domains such that the epitopes are not fully occluded.

We have also corrected phrasing at the end of this paragraph (line 270) to state that “Design-4 recapitulated a more desired inter-domain orientation.”

Line 408-9 is too strongly worded. STAC only strongly outperforms on I53-50, with perhaps

improvement (at one dilution point) for ferritin and none for liposomes. I think that the final conclusion should therefore be toned down a little.

The conclusion sentence has been updated to accurately summarize the results presented in this section and based on the statistical analyses performed. Specifically, we now state (lines 416–419):

Taken together, these data indicate that when formulated as protein nanoparticle immunogens, STAC can elicit a more potent functional response at higher sera dilutions when compared to Pfs230-D1 and Pfs48/45-D3, both as individual antigens and in combination as cocktails or mosaics.

Minor comments:

Figure 1c – could do with x axis labels for the black, yellow and blue circles.

This information has been newly incorporated in the figure legend.

Figure 1 – why not show binding data for the transmission blocking antibodies as well as the non-blockers which shouldn't bind?

Figure 1f-h demonstrates that certain Pfs230-D1 epitopes are masked in the context of a D2-containing Pfs230 construct and shows that while the Design-1 construct is partially able to mask these epitopes, it is not sufficient. Here, we have presented the binding that is most relevant to the design in Figure 1. The remaining data for representative antibodies with functional and non-functional TRA, which serve as controls that are not expected to change in affinity, are available to the reader in Supplementary figure 2 for the purpose of demonstrating the availability of other important epitopes for Design-1 compared to Pfs230-D1 and Pfs48/45-D3.

Throughout, it would help the reader to have a clearer explanation of what the different mAbs do/bind. More “Pfs230-binding mAb XXXX” etc...

We thank the Reviewer for this suggestion. To improve clarity, we have added Pfs230 and Pfs48/45 antibody descriptors to the text, particularly in sections related to Figures 1, 2, and 3.

Figure 4b, should that be STAC not design-22?

This has been corrected.

Figure 4f is missing x axis labels for the proteins.

This information has been newly incorporated in the figure legend.

Reviewer #2 (Remarks to the Author):

This paper generates a chimeric molecule combining two leading transmission-blocking vaccine candidates, Pfs230D1 and Pfs Pfs48/45-D3. These subdomains have been previously described to contain critical protective epitopes and have been used to elicit antibodies that block gametocyte ferritization in mosquitoes in complement-dependent (Pfs230D1) and complement-independent fashion, Pfs48/45. These candidate antigens are expressed on late-stage gametocytes, are immunogenic, and can boost antibody responses to vaccines based on these domains. This study uses a detailed structure study to generate a chimeric molecule that optimizes functional epitopes while masking epitopes that elicit non-functional Abs. They have successfully and convincingly accomplished this goal. The structure study and analysis well be done and clever. Overall comments on the structure studies initially described and followed by comments on the immunological and functional aspects.

We thank the Reviewer for their positive assessment of our work.

Comments related to the structural design

The authors of this study report a stabilized tandem antigen chimera of Pfs230-D1 (179 residues) and Pfs48/45-D3 (137 residues), representing the most potent epitopes of each antigen, thereby removing distracting epitopes from both. The individual constructs were combined with liposomes and used to immunize mice. They then examined the potential of linking both antigens in a single construct and evaluated, using BLI and known mAbs, the binding to the construct to confirm correctly folded domains and the masking of unwanted epitopes.

Design4 resulted in a crystal structure that is indistinguishable by rmsd from the native individual structures of Pfs230 and Pfs48/45, indicating correctly folded proteins despite linking them together.

In a next step, the authors applied AI-supported methods ProteinMPNN to improve the interface design and reduce unwanted binding, resulting in additional suitable designs. The improved designs increased expression yields and binding affinities to the tested mAbs. It would be interesting to note if the authors used the default weights for amino acids for their interface design or skewed towards residues typically found in interfaces.

We thank the Reviewer for their interest in the ProteinMPNN parameters used for this study. We have updated the Methods for the Computational design section (line 713) to indicate that we ran ProteinMPNN “*using the default weights for all twenty amino acids*”.

Paragraph 304-307

How many residues are involved in the interface in total? Are the 16 introduced mutations 20% of the interface, or is it more like 80%? This shows how dynamic an interface could be, redesigned while maintaining or improving the interface interaction so that others can derive their own interface designs for other proteins of interest.

We identified 34 residues at the interface which had sidechains which did or could project towards the linked domain. These residues were redesigned using ProteinMPNN. However, we pursued a somewhat conservative approach by selecting outputs with higher sequence recoveries to ensure proper domain folding. We have updated the Methods for the Computational design section to make this detail clear (lines 715–717):

Generated sequences were ranked by the ProteinMPNN sequence likelihood score with preference for higher sequence recovery to ensure proper domain folding and down selected to 15 sequences which were predicted with AlphaFold2.

Also there is a discrepancy in Figure 3c showing 18 mutated residues instead of 16?

We thank the Reviewer for catching this discrepancy. One stabilizing mutation on the Pfs48/45-D3 domain, N299Q, was missing and has been added to Figure 3c. The other mutation in Pfs230-D1, N585Q, is to remove a potential *N*-glycosylation site and is not found at the interface. N585Q is present in the clinical-stage Pfs230D1M-EPA candidate, as well as all the Pfs230-D1 constructs in this study. We have updated the Figure 3c caption to explain that N585Q is omitted for clarity.

9N8I

Chain A and E might be worth revisiting Molprobity to examine those atom clashes and potentially choose a different rotamer to improve the structure if the electron density supports that conformation.

This seems to be supported by section 6.1 with the rotamer outliers as well for Chain E. Also R_{free}/R_{work} seem to suggest a slight overfit, might be better to cut the resolution limit to 1.9 Å instead of 1.85 Å.

We thank the Reviewer for taking the time to inspect the validation reports of the structures presented in our manuscript. We had initially deposited a model with the anti-Kappa VHH sequence identity partially masked (using UNK pseudo-atoms in the model). We now incorporate the complete sequence and we have updated the model and validation report. With this change, the reported R_{free} value has improved from 0.259 to 0.234 and the RSRZ outliers have improved from 5.8% to 2.7%, while maintaining no Ramachandran or Sidechain outliers and a relatively low Clashscore. An update validation report is provided.

9N8N

Rotamers could be optimized.

Why did the authors model certain residues differently than the sequence construct suggests? I'm referring to the Table on Page 5, particularly residues 186, 195, 284, and 289.

Again it would be worth running Molprobity to optimize the rotamer outliers of Chain I

We have re-refined the molecular model of Design-4 bound to 2 potent Fabs. This allowed elimination of the two rotamer outliers and improved the RSRZ outliers from 6.8% to 5.8%, while

maintaining no Ramachandran outliers and a relatively low Clashscore and Rfree value. Regarding the sequences reported in the validation report, these are actually the differences between our engineered structure and the reference sequences of the native proteins displayed by Uniprot. We thank the Reviewer for flagging this potential issue. To be certain of the accuracy of our deposited models, we have double checked and confirmed that no sequence errors exist in any of the structures. For reference to the readers, we have included the amino acid sequences of the NF54 reference sequences for Pfs230 D1 and Pfs48/45 D3, as well as all mutations found in our relevant designed constructs in Supplementary figure 6d.

In addition to improvements in the molecular models determined by X-ray crystallography, we have also taken this opportunity to make minor improvements to the map and model from the cryo-EM dataset. All changes have been indicated in the Methods section Cryo-EM – Image processing. To this effect, the cFAR score has improved from 0.10 to 0.31, which has improved the backbone inclusion from 0.773 to 0.783. These changes also maintain no Ramachandran or Sidechain outliers, a low relative Clashscore and a favourable relative Q-score. All updated validation reports have been included and any relevant changes have been introduced in Supplementary tables 2 and 3, and Supplementary figures 7 and 8.

Minor

Line 175 define TRA

We have altered this section title at lines 177–178 to state “*Mixing Pfs230-D1 and Pfs48/45-D3 on adjuvanted liposomes elicits strong transmission-reducing activity*”. For reference, TRA remains defined in the Introduction at line 141.

Comments related to immunology and the functional characteristics of the vaccine design

The standard membrane feeding assay (SMFA) is used to evaluate IgG isotypes enriched from mice after vaccination with various constructs to measure the reduction in oocyte formation. (Do you think other isotypes might participate for protection?) These assays are well executed and are the typical functional tests to assess transmission-blocking activity. There was one biological replicate. The best chimeric molecule, STAC, shows comparable transmission-reducing activity (TRA) to the individual component antigens. In fact, its TRA is slightly lower than Pfs48/45 as a single molecule, suggesting that. There is only a slight improvement in TRA with STAC linked to nanoparticle immunization compared to the individual immunogens based on the SMFA model. Still, as the authors point out, the molecule is easy to produce and stable, which is important for vaccine candidate developability.

We thank the Reviewer for this positive assessment of our characterization of functional antibody responses to our nanoparticle immunogens. The SMFAs following immunizations with lipid nanoparticles and protein nanoparticles were performed by separate laboratories at the National Institutes of Health and Radboud University Medical Center, respectively, using distinct protocols (described in Methods). It has been previously shown that purified mouse polyclonal IgGs from Pfs230-D1 immunizations exhibited significantly greater SMFA activity in the

presence of human complement (Miura, et al., Vaccine. 2019). Therefore, for lipid and protein nanoparticle immunizations, mouse polyclonal antibodies were evaluated by SMFA in the presence of human complement proteins. For SMFAs following immunizations with lipid nanoparticles, IgG was purified from whole sera using Protein G affinity chromatography, so the effects of complement-fixing isotypes, like IgM, were not measured. In contrast, for SMFAs following immunization with protein nanoparticles, mouse sera were directly assessed by SMFA, again in the presence of human complement and this detail has been clarified for the reader in the Method section (lines 658–659) with the following sentence:

All SMFA experiments were conducted in the presence of active human complement.

The other aspect of the study was a vaccine molecule designed to mask the stimulation of non-functional antibodies. Theoretically, this is important, as a non-functional antibody could cause a steric hindrance to functional antibodies. Is there data to support this concerning Pfs230 or Pvs48/45? If so, this should be discussed or highlighted as a limitation.

For example if you add non-blocking mAbs to antibodies elicited by the different constructs do you observed alterations in TRA activity?

This is a very interesting point regarding both the quality of antibody responses, as well as the interplay of various monoclonals within a polyclonal mixture. As we describe in detail throughout the manuscript, our previous studies elucidating the epitopes on Pfs230-D1 and Pfs48/45-D3 targeted by antibodies from naturally infected individuals (Fabra-García, et al., 2023 and Ivanochko, et al., 2023) provided the rationale for the STAC concept. Importantly, non-functional antibodies directed to the buried D1-D2 interface on Pfs230 are only known to arise from immunizations with Pfs230-D1 subunit-based immunogens (also referenced in publications from the groups of P. Duffy and N. Tolia). We are not aware of studies that looked at mixing potent Pfs230 mAbs with these non-potent ones in SMFA, although such data is emerging for other malarial antigens such as RH5 where functional interplay has been demonstrated (Barrett, at al. Cell, 2025). To date, we are also not aware of any structural epitope mapping published for non-functional Pfs48/45-D3-directed mAbs, but fully agree that ongoing efforts to this effect will be valuable to address in the future. We have thus added this content to the Discussion (lines 451–461):

At present, a comprehensive epitope map of Pfs48/45-D3 is not available, with multiple surfaces unmapped for any known antibody epitope bins. Structural elucidation of full-length Pfs48/45 in complex with Pfs230 suggests certain epitopes on Pfs48/45-D3 may be occluded by Pfs230 in the heterodimer complex^{19,56}. These would become exposed in a Pfs48/45 domain-based subunit vaccine format, conceivably leading to the elicitation of non-functional antibodies directed to natively buried epitopes, as was the case for the Pfs230-D1 subunit vaccine. Moreover, given the limited clinical data available, it remains unclear whether Pfs230-D1 or Pfs48/45-D3 elicits the stronger functional immune response in malaria-endemic populations, where seroprevalence of antibodies to either Pfs230, Pfs48/45, or both antigens has been reported^{29,30}. This underscores the challenge of selecting an optimal TBV candidate, and

highlight the advantage of an approach that instead focuses on combining potent epitope bins present across both antigens.

Minor points

1. In Fig. 2, the relative binding shown in c and h can be better described. Is this relative binding of the new construct to the Pfs230 or Pfs48/45 individually? Please make this clearer.

Here, we present the binding signal relative to the maximum binding signal observed in the assays (for each specific antigen), which normalizes signal to 1 and enables the reader to see a relative rank ordering across both the Pfs230 and Pfs48/45 antigens of each designed molecule compared on an equivalent scale. We have also provided the absolute binding data for the reader in Supplementary figure 3a and 3c. The Methods section has been updated accordingly under the Biolayer interferometry – Quantitation section (lines 679–681):

To contrast the relative signals for each antigenic domain, relative binding signals were additionally calculated by normalizing each binding signal relative to the maximum observed binding signal for each specific antigen.

2. In some figures they used Design-22 and STAC interchangeably – this can be confusing. Please stick to one or the other designation as I presume these are the same constructs.

We have corrected the Figures and Figures legends for Figures 3 and 4, and Supplementary figures 6 and 7 to exclusively use ‘STAC’ after its definition on line 294.

3. Figure 5 is critical. The different color codes are difficult to follow, particularly with small figures with poor resolution. Suggest using different shapes and colors for the symbols denoting the different constructs.

We thank the Reviewer for recognizing the importance of Figure 5 and appreciate that the low resolution in the initial submission and similar symbols made it difficult to interpret this data. To mitigate this effect, this figure and all other figures have been provided at higher resolution and we have changed the symbols used for the I53-50-based immunogens (now depicted as triangles) to differentiate them from the ferritin-based immunogens (depicted as closed circles.)

Reviewer #3 (Remarks to the Author):

We appreciate this initiative from Nature Communications for recognizing the contributions from early career researchers and trainees during the peer review process and would also like to thank Reviewer 3 for taking time to review our manuscript.

Reviewer #1 (Remarks to the Author):

The authors have largely responded positively to my review and have made changes to the manuscript.

The one area in which I don't agree with their response relates to the depletion experiments. The aim of their overall study is to generate an immunogen which combines the best parts of Pfs48/45 and Pfs230 into a single immunogen which elicits inhibitory antibodies against both components. I therefore suggested a depletion experiment in which they deplete sera for Pfs230D1 and Pfs48.45 antibodies and see the contribution to SMFA, assessing whether they have achieved their objectives of raising both sets of antibodies. They have done this experiment and present it in the response to reviewers, but prefer to save it for a future paper.

The data is nice and makes it clear that TRA is almost entirely due to Pfs230D1 antibodies. This shows that the strategy to design an immunogen which elicits transmission blocking antibodies against both Pfs230 and Pfs48/45 has not been entirely successful as transmission blocking antibodies have not been elicited against Pfs48/45.

In the end, this is an editorial decision. However, my view is that this data should be included and discussed in the paper as it directly relates to whether the aims of this study have been achieved.

We thank Reviewer 1 for their appreciation of the additional data shared during revisions. Given the Editor also viewed its inclusion favorably, the data has now been incorporated in the Results section and in Supplementary figure 10c-d. As requested, the implications of our findings are also further discussed in the Discussion section: "By mitigating non-functional antibody interference, STAC may overcome limitations of leading TBV candidates and represent an important epitope-masking strategy with broader applications in vaccine development. Importantly, the genetic conjugation of distinct epitopes may confer immunodominance hierarchies between epitopes, as we have observed in rodent immunizations with STAC which elicits a Pfs230-D1-dominant response. Several factors may contribute to this phenomenon, including germline-encoded bias of specific B cell lineages for residue signatures at specific epitopes⁷⁰, proteolysis sensitivity in B cell follicles for specific epitopes⁷¹ or epitope prominence by antigen orientations on nanoparticles⁷². Investigating the rules of immunodominance for different epitopes, and their contributions to function across preclinical species and in the context of the human Ig repertoire will be important next steps to further understand responses to STAC-based immunogens, and whether further engineering is warranted to obtain a more balanced response to the potent Pfs230 and Pfs48/45 epitopes in a human Ig repertoire context."

Reviewer #2 (Remarks to the Author):

The authors have adequately address our concerns

We thank Reviewer 2 for taking time to review our manuscript and for their favorable assessment of our work.

Reviewer #3 (Remarks to the Author):

We appreciate this initiative from Nature Communications for recognizing the contributions from early career researchers and trainees during the peer review process and would also like to thank Reviewer 3 for taking time to review our manuscript.